# Ammonia marine engine design for enhanced efficiency and reduced greenhouse gas emissions

Xinyi Zhou [1,2,3], Tie Li [1,2] ✉, Run Chen [1,2], Yijie Wei[4], Xinran Wang[2], Ning Wang[2], Shiyan Li[1,2], Min Kuang[5] & Wenming Yang [3] ✉

Pilot-diesel-ignition ammonia combustion engines have attracted widespread attentions from the maritime sector, but there are still bottleneck problems such as high unburned $NH_3$ and $N_2O$ emissions as well as low thermal efficiency that need to be solved before further applications. In this study, a concept termed as in-cylinder reforming gas recirculation is initiated to simultaneously improve the thermal efficiency and reduce the unburned $NH_3$, $NO_x$, $N_2O$ and greenhouse gas emissions of pilot-diesel-ignition ammonia combustion engine. For this concept, one cylinder of the multi-cylinder engine operates rich of stoichiometric and the excess ammonia in the cylinder is partially decomposed into hydrogen, then the exhaust of this dedicated reforming cylinder is recirculated into the other cylinders and therefore the advantages of hydrogen-enriched combustion and exhaust gas recirculation can be combined. The results show that at 3% diesel energetic ratio and 1000 rpm, the engine can increase the indicated thermal efficiency by 15.8% and reduce the unburned $NH_3$ by 89.3%, $N_2O$ by 91.2% compared to the base/traditional ammonia engine without the proposed method. At the same time, it is able to reduce carbon footprint by 97.0% and greenhouse gases by 94.0% compared to the traditional pure diesel mode.

According to the latest definition in 2023 by Lloyd's Register[1], "zero greenhouse gas (GHG) emissions in the maritime sector" can be divided into (1) Absolute-zero emissions mean that there are no GHG emissions from the whole Well-to-Wake lifecycle; (2) Net-zero emissions can be achieved when anthropogenic GHG emissions to atmosphere can be balanced by anthropogenic removal from the perspective of Well-to-Wake lifecycle; (3) Near-zero emissions refer to a reduction of more than 80% in GHG emissions compared to low-sulfur fuel oil. On July 7, 2023, the International Maritime Organization (IMO) proposed a reduction of at least 70% in GHG from international ships by 2040, with the ultimate goal of achieving net-zero GHG shipping by around 2050[2]. As a result, replacing the marine oil-fueled diesel engines, which account for around 98.8% of the prime movers in international ships[3] with alternative fuel-fueled engines that feature near-zero or net-zero GHG emissions is inevitable to achieve the IMO target in the next 30 years.

Ammonia, hydrogen, methanol, and methane have been considered as potential alternative fuels for the future maritime sector[4,5]. According to the fuel production routes, these alternative fuels are generally divided into (1) Gray/Brown fuels, utilizing natural gas/coal as feedstocks, and no carbon capture technology is used in the fuel production pathway; (2) Blue fuels, still using fossil fuel as feedstocks,

[1]State Key Laboratory of Ocean Engineering, Shanghai Jiao Tong University, Shanghai, People's Republic of China. [2]Institute of Power Plants and Automation, Shanghai Jiao Tong University, Shanghai, People's Republic of China. [3]Department of Mechanical Engineering, National University of Singapore, Singapore, Singapore. [4]National Engineering Research Center of Special Equipment and Power System for Ship and Marine Engineering, Shanghai, People's Republic of China. [5]Faculty of Maritime and Transportation, Ningbo University, Zhejiang, People's Republic of China. ✉e-mail: litie@sjtu.edu.cn; mpeywm@nus.edu.sg

but the carbon dioxide ($CO_2$) emissions during the fuel production processes are captured and permanently stored; (3) Green fuels that are produced from renewable energy sources such as wind or solar power. The IMO highlights that decarbonizing global shipping should not shift emissions to other sectors[3]. As a result, although most of these alternative fuels produced to date are Gray and Brown fuels, the maritime sector primarily focuses on the use of Blue/Green fuels that have the potential of near-zero/net-zero Well-to-Wake GHG emissions[3–5]. It should be noted that if the focus is upgraded to absolute-zero emissions and the onboard carbon capture is not used, any fuel that contains carbon such as methanol and methane should be precluded, which means that only the fuels that contain no carbon like hydrogen and ammonia can be considered as candidates.

Recently, there have been many studies comparing the application of various alternative fuels in the marine sector. For example, a recent analysis paper in Nature Energy[6] compared various alternative marine fuels such as hydrogen, ammonia, methane, and methanol. It reported that ammonia is the most balanced carbon-free fuel, while methanol is the most balanced carbonaceous fuel. Another paper in Renewable and Sustainable Energy Reviews[7] briefly compared the use of ammonia, methanol, and hydrogen in the maritime sector. It reported that owing to the low storage and transportation costs and well-established infrastructures, ammonia is an ideal alternative fuel for future international shipping. Kanchiralla et al.[8] compared the lifecycle costs of using hydrogen, ammonia, and methanol as alternative fuels in three background ships. They found that the use of ammonia fuel has the lowest lifecycle cost for all three background ships. Besides the economic advantages mentioned in[6,8], when compared to carbon-containing fuels like methanol and methane, the combustion of carbon-free ammonia fuel produces no engine-out emissions such as $CO_2$, particulate matter, carbon monoxide (CO), hydrocarbon, formaldehyde, and so on. Moreover, the synthesis of ammonia does not require the expensive direct air capture or bioenergy with carbon capture and storage processes, which are necessary for obtaining reasonable sources of $CO_2$ during the fuel production processes of methanol and methane[4,9]. Furthermore, the use of ammonia in the maritime sector need not specially establish and monitor the complex carbon value chain compared to the use of methanol and methane fuels. Currently, in addition to the shortage of Blue and Green sources that all alternative fuels face, the most significant challenge in using ammonia as a marine fuel lies in the poor combustion characteristics of ammonia and the relevant engine combustion technologies[10,11].

As discussed in the above paragraphs, ammonia is an ideal alternative fuel for the maritime sector, and the most significant challenge in using ammonia as a marine fuel lies in engine combustion technology. To overcome ammonia's inherent shortcoming of high ignition energy, narrow flammability limit, and slow propagation speed[10,11], various pilot-diesel-ignition ammonia combustion modes have received widespread attention[12,13]. Of these, high-pressure injection dual-fuel (HPDF) and low-pressure injection dual-fuel (LPDF) are two major modes[7,14]. For the former mode, the liquid-phase ammonia is directly injected into the cylinder near the completion of the compression stroke and ignited by high-reactivity diesel fuel. For the latter one, the ammonia is supplied into the intake manifold and introduced into the cylinder together with the intake air during the intake stroke, and then diesel is injected into the cylinder to ignite the ammonia-air mixture by the end of the compression stroke. The HPDF mode has the potential to mitigate the unburned $NH_3$ in the clearance volume and overcome the fuel slip during the valve overlap[15], and it has attracted many attempts recently[16,17]. However, the unburned ammonia emissions in the latest experimental results of the HPDF mode are not satisfactory. For example, Scharl et al.[18] reported that the liquid ammonia spray flame is difficult to stabilize after diesel ignition in an optical experiment, which results in approximately 10% to 15% of the

ammonia fuel not being burned if additional post-diesel injection is not employed[19]. The only available experimental data about unburned ammonia emissions from HPDF engine tests is provided by collaborative research involving institutions such as MAN Energy Solutions, Technical University of Munich, Woodward L'Orange, Neptun Ship Design, and others[20]. They exhibited that the unburned ammonia emissions are around 6800 ppm and 9300 ppm at ammonia energetic ratio of 50% and 90%, respectively, using a single-cylinder engine with 175 mm bore diameter, 19:1 compression ratio, 1800 rpm engine speed, 40 MPa liquid ammonia injection pressure, and 17 bar indicated mean effective pressure (IMEP). In addition to the unacceptable high unburned ammonia emissions, owing to the corrosive nature of ammonia to brass, copper, and zinc alloys, it is still expensive and challenging to design a proper liquid ammonia supply and injection system for large-scale commercial applications of the HPDF mode. Moreover, placing additional liquid ammonia injectors on the already compact cylinder head also poses challenges for cylinder head design.

Compared to the HPDF mode, the LPDF mode requires fewer modifications to the original diesel engine and has attracted increasing interest in recent years. Moreover, Zhou et al.[7] and Li et al.[14] reported that the LPDF has the potential to achieve a better fuel economy compared to the HPDF mode owing to the reduced heat transfer, which makes the LPDF mode more attractive. However, according to the latest experimental results during the past two years, the LPDF mode also suffers from the major issue of high unburned ammonia and $N_2O$ emissions. For example, in 2022, Yousef et al.[21] studied the effects of diesel injection timing on combustion characteristics of the LPDF mode using an engine with a 137.2 mm bore diameter. At the operating condition of 910 rpm, 8.1 bar IMEP, and 40% ammonia energetic ratio, they found that the lowest unburned ammonia and $N_2O$ emissions in the interested diesel injection timing range are 15.5 g/kWh (or 4445 ppm) and 0.6 g/kWh, respectively. In 2023, Nadimi et al.[22] studied the effects of ammonia energetic ratio on emissions and engine performance of the LPDF mode using an 86 mm bore diameter engine under full load and 1200 rpm. They reported that the unburned ammonia increases from approximately 4000 ppm to 14,500 ppm when the ammonia energetic ratio increases from 14.9% to 84.2%. The effects of ammonia energetic ratio on unburned ammonia and $N_2O$ emissions were also studied by Jin et al.[23], and the LPDF engine with 116 mm bore diameter was running at 1000 rpm and 6 bar IMEP. They reported that at 50% ammonia energetic ratio, the unburned ammonia and $N_2O$ emissions were 31 g/kWh and 1.12 g/kWh, respectively, while at an increased ammonia energetic ratio of 90%, the unburned ammonia and $N_2O$ will increase to 143 g/kWh and 1.75 g/kWh, respectively. Since $N_2O$ has a greenhouse effect around 265–298 times that of $CO_2$, the $N_2O$ emissions at this level will significantly offset the beneficial effect of GHG reduction by ammonia-fueled engines and could deteriorate the global nitrogen cycle as highlighted in the latest comment in Nature Energy[24].

Recently, the traditional strategies such as split diesel injection[25–27] and very early diesel injection[28], which are commonly used in combustion optimization of diesel engines, have received much attention to address the issues of high unburned ammonia emissions. With the split diesel injection strategies, since part of the high-reactivity diesel fuel (i.e., pilot fuel) has been fully mixed with the ammonia-air mixture during the compression stroke, the chemical reaction activity of the combustible mixture is improved and therefore the unburned ammonia emissions can be effectively reduced. For example, in 2023, Mi et al.[29] investigated the potential of pre-main diesel injection strategy on unburned $NH_3$ reduction using a 114 mm bore diameter LPDF engine. They found that the proper use of the pre-main injection strategy can reduce the unburned ammonia from around 8700 ppm to 4400 ppm under the operating condition of 1500 rpm, 10 bar IMEP, and 70% ammonia energetic ratio. However, the increased chemical

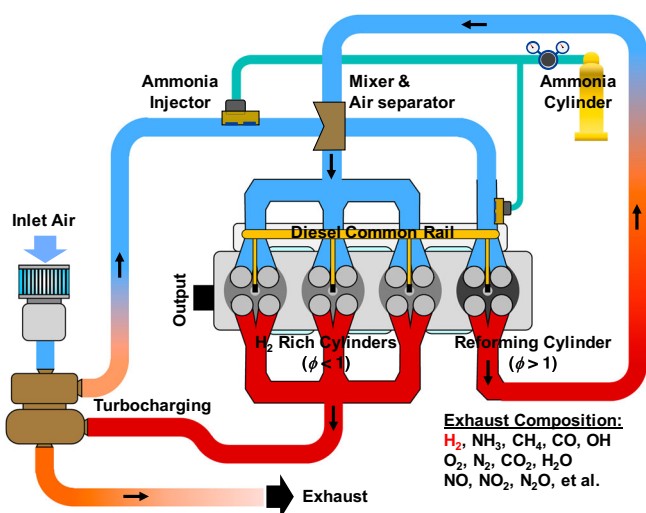

**Fig. 1 | Conceptual model of the in-cylinder reforming gas recirculation (IRGR) for pilot-diesel-ignition ammonia combustion engine.** Here, $\phi$ refers to the overall equivalence ratio.

reaction activity of the combustible mixture by diesel pre-injection also significantly increases the combustion temperature and $NO_x$ emissions, leading to trade-off relationships between $NO_x$ and unburned ammonia emissions. Moreover, under the above operating conditions, the experimental results exhibited that the pre-main injection strategy will also increase the $N_2O$ emissions, which offsets the benefits of greenhouse gas emission reduction. Recently, Shin et al.[28] exhibited a numerical simulation to clarify the potential of very early diesel injection on unburned ammonia reduction and fuel economy improvement of the LPDF mode. With a 137 mm bore diameter engine running at 910 rpm, they found that the very early diesel injection at −80°CA after top dead center (aTDC) can obviously reduce the unburned ammonia, but also reported that the NO emissions will increase from around 1500 ppm to 6000 ppm when the diesel injection timing is advanced from −15° to −80°CA aTDC, which once again demonstrated the trade-off relationship between unburned ammonia and $NO_x$ emissions.

Obviously, the latest results obtained from various engine types and operating conditions indicate that the high unburned ammonia and $N_2O$ emissions are critical challenges hindering the widespread application of pilot-diesel-ignition ammonia combustion engines. Technologies are urgently needed to solve bottleneck problems such as high unburned ammonia and $N_2O$ emissions, the trade-off relationship between unburned ammonia and $NO_x$ emissions, and low efficiency. It is well-known that hydrogen addition is a useful method to improve the combustion efficiency of ammonia[30,31], while using exhaust gas recirculation (EGR) to control $NO_x$ emission is a common approach for traditional diesel and gasoline engines[32,33]. In light of this, a concept termed in-cylinder reforming gas recirculation (IRGR) is initiated in the present study to simultaneously improve the indicated thermal efficiency and reduce the unburned $NH_3$, $NO_x$, $N_2O$, and GHG emissions of pilot-diesel-ignition ammonia combustion engine. As shown in Fig. 1, for this concept, one cylinder of the multicylinder engine operates rich in stoichiometric and the excess ammonia is partially decomposed into hydrogen under the incylinder high-temperature environments, then the exhaust of this dedicated reforming cylinder is recirculated into the other cylinders and therefore the advantages of hydrogen-enriched combustion and EGR can be combined. Compared with the approach of equipping an additional reformer outside the engine, the concept in this study has the following advantages: (1) saving around 23% of the fuel energy due to the heat transfer loss of the external reformer[34,35]; (2) solving

the problem of short life and high cost of catalyst; (3) avoiding the complexity of engine structure. It should be noted that a comparable technology termed dedicated EGR was proposed by Alger et al.[35] in 2007 to improve the EGR tolerance of a spark-ignition gasoline engine. The initial version of dedicated EGR[35] includes a water-gas shift catalyst in the exhaust manifold to promote the water-gas shift reaction (i.e., $CO + H_2O = CO_2 + H_2$), it was found that a small amount of hydrogen (i.e., around 0.2% by volume) added in the intake manifold is helpful for improving the combustion efficiency and engine performance. Two years later, Alger et al.[36] proposed a modified version of dedicated EGR in 2009, one of the main modifications is the use of a partial oxidation catalyst in the exhaust runner of the dedicated cylinder to convert some of the unburned hydrocarbons in the exhaust to CO and $H_2$. They reported that the dedicated EGR is useful for reducing fuel consumption and improving engine stability. In 2014, Chadwell et al.[37] further improved the dedicated EGR technology by optimizing the relevant boosting, EGR control, EGR mixing, and ignition systems, and obtained the performance map of a 2.0 L gasoline direct injection engine. They demonstrated that the dedicated EGR technology can improve the engine efficiency by around 10%. Later, Gukelberger et al.[38,39] further studied the potential of applying dedicated EGR to a gasoline engine and evaluated the engine performance under various engine loads. They demonstrated that the engine with dedicated EGR has the ability to operate at a high compression ratio and therefore has higher thermal efficiency. Moreover, they highlighted that a watergas shift catalyst added in the exhaust runner of the dedicated cylinder can promote the water-gas shift reaction and increase the EGR quality. In recent years, there have also been some studies demonstrating that the dedicated EGR can improve the dilution tolerance of spark-ignition natural gas engine[40]. Compared to the above dedicated EGR technology proposed for improving the EGR tolerance of spark-ignition engine, the IRGR pilot-diesel-ignition ammonia combustion engine in the present study exhibits differences in terms of the independence of Cylinder #1's intake system (i.e., independent for IRGR vs shared with other cylinders for dedicated EGR), whether the EGR catalyst is suggested, ignition and combustion mode, fuels (i.e., ammonia for IRGR vs. carbonaceous fuels like gasoline and natural gas for dedicated EGR) and so on. Especially, so far there has been no study to verify the potential of a similar concept applied to ammonia-fueled engines.

In the present study, a chemical reaction mechanism for the combined ammonia and n-heptane combustion is developed and integrated into the numerical simulation platform, and a four-cylinder pilot-diesel-ignition ammonia dual-fuel combustion engine without IRGR system was used to obtain the experimental data for validating the numerical models and chemical kinetic mechanism, as detailed in the section of 'Methods'. Then, the potential of the IRGR concept on thermal efficiency improvement and emission reduction is numerically studied under various ammonia energetic ratios (i.e., 80%, 90%, 97%), engine speeds (i.e., 1000, 1500, 2000, 2500 rpm), fuel injection timings (i.e., from −20 to 0°CA aTDC) and fuel enrichment levels of the dedicated reforming cylinder. Lastly, the primary results are discussed and the findings are summarized.

## Results

### Dedicated reforming cylinder of IRGR engine

Figure 2a exhibits the evolutions of in-cylinder pressure and temperature as well as the heat release rate of the dedicated reforming cylinder under various overall excess air ratios, while Fig. 2d shows the development of the high-temperature region and the hydrogen-rich region as well as in-cylinder temperature distributions of the dedicated reforming cylinder. In Fig. 2d, the red, green, and cyan isosurfaces indicate a temperature of 2000 K, a hydrogen mole fraction of 4.5% and 9.0%, respectively. For the dedicated reforming cylinder, the

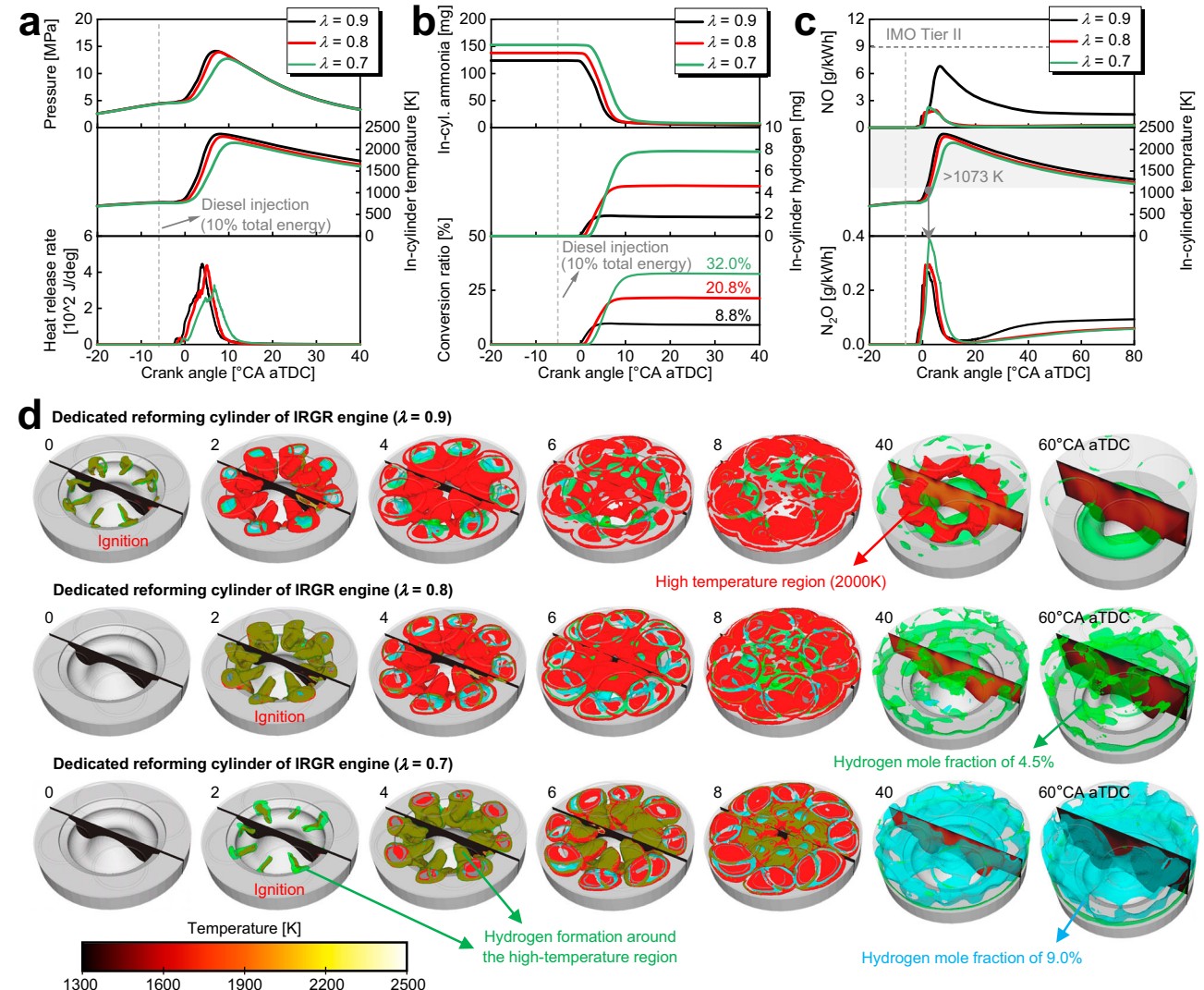

**Fig. 2 | Combustion and emission characteristics of the dedicated reforming cylinder under various overall excess air ratios. a** In-cylinder temperature and pressure as well as heat release rate, **b** In-cylinder ammonia and hydrogen mass as well as hydrogen conversion ratio, **c** In-cylinder NO and $N_2O$, **d** Development of high-temperature region and hydrogen-rich region as well as in-cylinder temperature distributions at different crank angles. Here, $\lambda$ refers to the overall excess air ratio. In **b**, the hydrogen conversion ratio is calculated as the ratio of in-cylinder hydrogen energy and total input ammonia energy. In **d**, the red, green, and cyan isosurfaces indicate the temperature of 200 K, and hydrogen mole fraction of 4.5% and 9.0%, respectively. (The following remain consistent across all cases: 1000 rpm, 10% diesel energetic ratio, −6°CA aTDC diesel injection, and 318 K intake temperature). Source data are provided as a Source Data file.

overall excess air ratios concerned here are 0.7 and 0.8, and the 0.9 overall excess air ratio case is also exhibited here for comparative purposes. Moreover, the following remain consistent across all cases: 1000 rpm, 318 K intake temperature, 10% diesel energetic ratio, and −6°CA aTDC injection. Observing the heat release profiles in Fig. 2a and the high-temperature region in Fig. 2d, it is found that the higher the fuel enrichment level, the longer the ignition delay. Since the combustion phasing is delayed, the case with the higher fuel enrichment level features a reduced in-cylinder maximum pressure and temperature. Careful observation of the picture in Fig. 2d shows that the hydrogen is mainly generated near the high-temperature region, where the reactions of hydrogen production by ammonia decomposition and reforming (i.e., $NH_3 + H = NH_2 + H_2$ and $NH_3 + M = NH_2 + H + M$) are most positive, and the higher the fuel enrichment level, the more hydrogen is generated. For example, the area of 9.0% hydrogen mole fraction isosurface highlighted by cyan color is negligibly small for the 0.8 and 0.9 overall excess air ratio cases, while the whole cylinder is filled with a high concentration of reformed and decomposed hydrogen at 40 and 60°CA aTDC for the

0.7 overall excess air ratio case. Figure 2b shows the evolutions of in-cylinder ammonia and hydrogen mass as well as the hydrogen conversion ratio of the dedicated reforming cylinder. Here, the hydrogen conversion ratio is calculated as the ratio of in-cylinder hydrogen energy and total input ammonia energy. Since all the cases keep the constant intake temperature and pressure, the initial in-cylinder ammonia mass increases with the fuel enrichment level increasing. For the 0.7, 0.8, and 0.9 overall excess air ratio cases, approximately 7.7 mg, 4.5 mg, and 1.7 mg of hydrogen are generated during the ammonia combustion processes, respectively, corresponding to hydrogen conversion ratios of 32.0%, 20.8%, and 8.8%, respectively. It should be noted that for the real operation of an IRGR ammonia combustion engine, it is necessary to refer to the experiences of hydrogen engine design to avoid the occurrence of backfire.

Figure 2c exhibits the in-cylinder NO and $N_2O$ evolutions of dedicated reforming cylinders under various overall excess air ratios. The overall excess air ratios concerned here are 0.7 and 0.8, and the 0.9 overall excess air ratio case is also included here for comparison. For the ammonia combustion engine, except for the formation of

thermal NO, the fuel NO is also inevitably formed during the combustion processes because the ammonia fuel contains the N atom itself. As a result, the in-cylinder NO increases rapidly after the start of combustion. It is well-known that the condition for the large amount of NO formation is high-temperature and oxygen-rich environments. Owing to the relatively higher in-cylinder temperature and oxygen concentration, the 0.9 excess air ratio case exhibits the highest NO formation. Different from diesel engines where in-cylinder NO reaches a peak and then remains constant[41], for the ammonia-fueled engine here, the in-cylinder NO reaches a peak and then go through a decreasing phase prior to reaching the quasi-steady state. This can be attributed to the thermal DeNO processes[42], whose main reactions are $NO + NH_2 = NNH + OH$ and $NO + NH_2 = N_2 + H_2O$[43]. Generally, the NO concentration at the exhaust valve open timing of the dedicated reforming cylinder is significantly lower than the IMO Tier II limit. The in-cylinder $N_2O$ also increases rapidly after the combustion startup, and this is because the $N_2O$ production reactions are initiated as soon as the ammonia combustion starts, and mainly include the following two reactions[44,45]: $NH + NO = N_2O + H$ and $NH_2 + NO_2 = N_2O + H_2O$. Since the in-cylinder $N_2O$ will be thermally decomposed above 1073-1273 K[46], the moment when the in-cylinder temperature reaches 1073 K is indicated in the plot of $N_2O$ evolutions. It can be seen that the moment when $N_2O$ begins to decrease is almost overlapped with the time when the in-cylinder temperature exceeds 1073 K. It should be noted that the $N_2O$ begins to show a rising trend again after around 20°CA aTDC. This is because the $N_2O$ production reactions become more dominant compared to the thermal decomposition reactions owing to the decreased in-cylinder temperature at this stage. According to recent studies about ammonia spark-ignition engines[47], adding hydrogen with a small fraction can obviously improve thermal efficiency and reduce unburned ammonia. Figure 2b shows that the hydrogen conversion ratio of the dedicated reforming cylinder under the concerned fuel enrichment levels is around 20.8% to 32.0%. In the following sections, the exhaust of the dedicated reforming cylinder including $H_2$, $NH_3$, CO, $O_2$, $N_2$, $CO_2$, $CH_4$, NO, $NO_2$, $N_2O$, etc. will be recirculated to Cylinders #2–4, and the engine performance and emissions of the base engine and IRGR engine will be compared and discussed under various diesel energetic ratios (i.e., 3%, 10%, and 20%), fuel enrichment level of Cylinder #1 (i.e., 0.7 and 0.8), injection timings (i.e., from −20 to 0°CA aTDC), and engine speeds (1000, 1500, 2000 and 2500 rpm).

## Comparison of the IRGR engine and base engine

For the IRGR engine, most cylinders (i.e., Cylinders #2–4) operate under the hydrogen enrichment condition, and the combustion characteristics of these cylinders dominate the performance of the entire engine. Figure 3a gives the comparison of in-cylinder temperature and pressure as well as heat release rate between the base engine without the IRGR system and Cylinders #2–4 of the IRGR engine. The following are constant for direct comparison: 1717 J total input energy per cylinder, 10% diesel energetic ratio, 1000 rpm, and −6°CA aTDC diesel injection timing. The overall excess air ratio of the base engine without the IRGR system is 1.5, while considering the reforming gas recirculation from Cylinder #1, the value of Cylinders #2–4 of the IRGR engine is reduced to 1.1. Here, the combustion characteristics of Cylinders #2–4 of the IRGR engine are evaluated under different fuel enrichment levels of Cylinder #1 (i.e., 0.7 and 0.8 overall excess air ratios). It can be seen that the ignition delay and combustion duration are similar for the base engine and Cylinders #2–4 of the IRGR engine ($\lambda_{Cyl. \#1} = 0.8$). This is because $H_2$, $NH_3$, CO, $N_2$, $CO_2$, CO, $CH_4$, NO, $NO_2$, $N_2O$, etc. from the exhaust of the dedicated reforming cylinder are all recycled to Cylinders #2–4, and the combustion promotion by hydrogen addition and combustion inhibition by introduced $CO_2$ and $N_2$ are offsetting each other in Cylinders #2–4. For Cylinders #2–4 of the IRGR engine ($\lambda_{Cyl. \#1} = 0.7$), the increased hydrogen enrichment level will lead to a higher heat release rate and a slightly shortened

combustion duration, leading to an increased in-cylinder peak pressure and temperature.

Figure 3b gives the comparison of flame evolution and ammonia mass fraction distribution between the base engine without the IRGR system and Cylinders #2–4 of the IRGR engine. The operating condition is consistent with Fig. 3a. Here, the temperature information is overlaid on the isosurfaces of 1.1 excess air ratio, and the cross-sectional planes are colored with ammonia mass fraction. Generally, the flame temperature of Cylinder #2–4 of the IRGR engine is lower than that of the base engine, which can be attributed to the increased $CO_2$ of the former case. Upon careful observation of the ammonia mass fraction slices at 14°CA aTDC, it can be seen that there are many unburned $NH_3$ in the clearance between the piston and cylinder head of the base engine, while there is almost no unburned ammonia in the clearance of Cylinders #2–4 of the IRGR engine ($\lambda_{Cyl. \#1} = 0.7$). This is because the hydrogen addition increases the chemical reaction activity of the combustible mixture and reduces the quenching distance. Generally, the concept initiated in this study can significantly reduce the unburned ammonia of the pilot-diesel-ignition ammonia combustion engine, and more details about the emission reduction will be given in the next paragraphs.

Figure 3c exhibits the comparison of unburned $NH_3$, $NO_x$, and $N_2O$ emissions between the base/traditional ammonia engine without an IRGR system and the IRGR ammonia engine, while Fig. 3d gives the comparison of heat balance. Here, the object of comparison is the entire engine including all four cylinders. The following remain consistent for all the engine types and cylinders: diesel energetic ratio of 10%, injection timing of −6°CA aTDC, 1000 rpm, and 318 K intake temperature. Moreover, for the base engine and Cylinders #2–4 of the IRGR engine, the total input energy per cylinder is constant at 1717 J. Figure 3c shows that the IRGR concept can reduce approximately 84% of the unburned ammonia, mainly because the hydrogen addition increases the chemical reaction activity of the combustible mixture and significantly reduces the unburned ammonia in the clearance volume, as discussed in Fig. 3b. The $N_2O$ emissions are significantly reduced for the IRGR engine, and reasons can be attributed to the reduced oxygen content and the suppressed $N_2O$ production reactions. It can be seen that the $NO_x$ emissions of the IRGR ammonia engine are also obviously smaller than those of the base/traditional ammonia engine. Firstly, this is because of the reduced oxygen content of the IRGR engine, which leads to a decrease in the formation of thermal $NO_x$. Moreover, for Cylinders #2–4 of the IRGR engine, a small portion of the energy originally provided by ammonia is replaced by hydrogen recirculated from Cylinder #1, which helps to reduce the formation of fuel $NO_x$. It should be noted that the reduction of $NO_x$ emissions by the IRGR concept decreases at the higher fuel enrichment level of Cylinder #1. This is because more hydrogen is recirculated from Cylinder #1 to Cylinders #2–4 at the increased fuel enrichment level, which leads to the formation of more thermal $NO_x$. Figure 3d shows that the unburned loss is reduced from 11.6% to around 3.0% when employing the IRGR technology, and this is primarily due to the reduction of unburned ammonia in the clearance volume. The cooling losses of the IRGR engine are higher than that of the base engine, and this can be attributed to the higher in-cylinder temperature by improved combustion of the IRGR engine, as shown in Fig. 3a. Generally, this concept is helpful for improving the fuel economy owing to the significant reduction in unburned losses.

## The potential of IRGR under different operating conditions

Figure 3 exhibits that the IRGR concept is an effective method to simultaneously improve the thermal efficiency and reduce the unburned ammonia, $NO_x$, and $N_2O$ emissions under 1000 rpm, 10% diesel energetic ratio, and −6°CA aTDC fuel injection timing. In this section, the effects of IRGR technology on thermal efficiency improvement and emission reduction will be studied under more

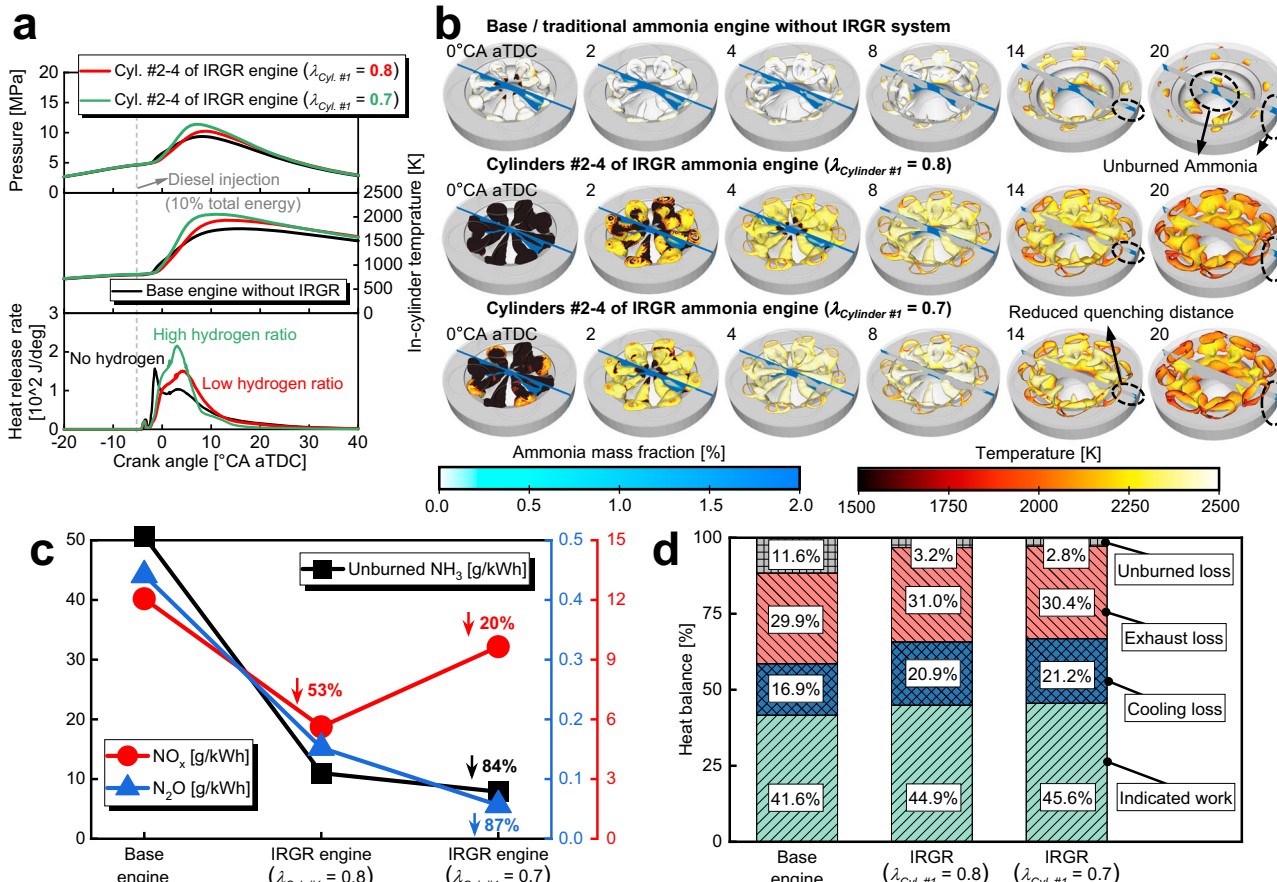

**Fig. 3 | Comparison between the base/traditional ammonia engine without IRGR system and IRGR ammonia engine. a** Comparison of in-cylinder temperature and pressure as well as heat release rate between the base/traditional ammonia engine without IRGR system and Cylinders #2–4 of IRGR ammonia engine, **b** Comparison of flame evolution and ammonia mass fraction distribution between the base/traditional ammonia engine without IRGR system and Cylinders #2–4 of IRGR ammonia engine at different crank angles, **c** Comparison of unburned NH₃, NOₓ and N₂O emissions between the base/traditional ammonia engine without IRGR system and IRGR ammonia engine, **d** Comparison of heat balance between the base/ traditional ammonia engine without IRGR system and IRGR ammonia engine. Here, $\lambda$ refers to the overall excess air ratio. In **b**, the temperature information is overlaid on the isosurfaces of 1.1 excess air ratio, and the cross-sectional planes are colored with ammonia mass fraction. In **c**, **d**, the object of comparison is the entire engine including all four cylinders. (The following remain consistent for all the engine types and cylinders: diesel energetic ratio of 10%, 1000 rpm, intake temperature of 318 K, and diesel injection timing of −6°CA aTDC; For the base/traditional ammonia engine and Cylinders #2–4 of the IRGR ammonia engine, the total input energy per cylinder is constant at 1717 J). Source data are provided as a Source Data file.

ammonia energetic ratios and a wide range of fuel injection timings to fully explore the potential of IRGR concept, as shown in Fig. 4. For fair comparison, the object of comparison is the entire engine including all the four cylinders, important parameters such as 1000 rpm and 318 K intake temperature are kept constant for all the engine types and cylinders, and the total input energy per cylinder is constant at 1717 J for the base engine and Cylinders #2–4 of the IRGR engine. Figure 4a–c gives the comparison of NOₓ emissions between the base/traditional ammonia engine without IRGR system and IRGR ammonia engine under ammonia energetic ratio of 80%, 90%, and 97%, and the IMO Tier II NOₓ limit is shown in the figure to highlight the interest range of fuel injection timings. For most cases, the IRGR technology can reduce the NOₓ emissions owing to the reduced oxygen content by reforming gas recirculation effects, thereby extending the useful range of fuel injection timing. Figure 4d–f shows the comparison of indicated thermal efficiency, while Fig. 4g–i gives the comparison of unburned NH₃ emissions. It is found that the advanced pilot-diesel injection within a certain range aids in the reduction of unburned ammonia emissions, and this can be attributed to the increased timescale for diesel and ammonia mixing and chemical reaction at the early injection timing. The indicated thermal efficiency gradually increases as the diesel injection timing is advanced within a certain range, mainly because more heat is

released near the TDC and the increased combustion efficiency of ammonia. However, further advancing the injection timing will reduce the degree of constant volume heat release and lead to a deteriorated fuel economy. Since the unburned ammonia has already exceeded 72 g/kWh, the indicated thermal efficiency of base/traditional ammonia engine without IRGR system is correspondingly reduced to below 40% at 97% ammonia energetic ratio. Fortunately, the IRGR concept remains highly effective even at 97% ammonia energetic ratio, as it can reduce unburned ammonia to around 8 g/kWh and increase the indicated thermal efficiency to around 46%. The CO₂ emission reduction and ammonia energetic ratio increase are positively correlated. However, for the base/traditional ammonia engine without an IRGR system, when the ammonia energy ratio increases from 80% to 97%, the N₂O emissions also increase from approximately 0.5 to 0.8 g/kWh. Since N₂O has a greenhouse effect around 298 times that of CO₂[24], the N₂O emissions at this level will lead to a GHG effect comparable to the CO₂ emissions from the pure diesel mode (i.e., around 600 g/kWh), and more details about the potential of IRGR concept on GHG reduction will be given in the following 'IRGR technology for GHG reduction' section.

In Fig. 4, the best indicated thermal efficiency points within the IMO Tier II NOₓ limit are highlighted by the dashed circles, and these best points are compared in Fig. 4m–o. For the IRGR engine, more

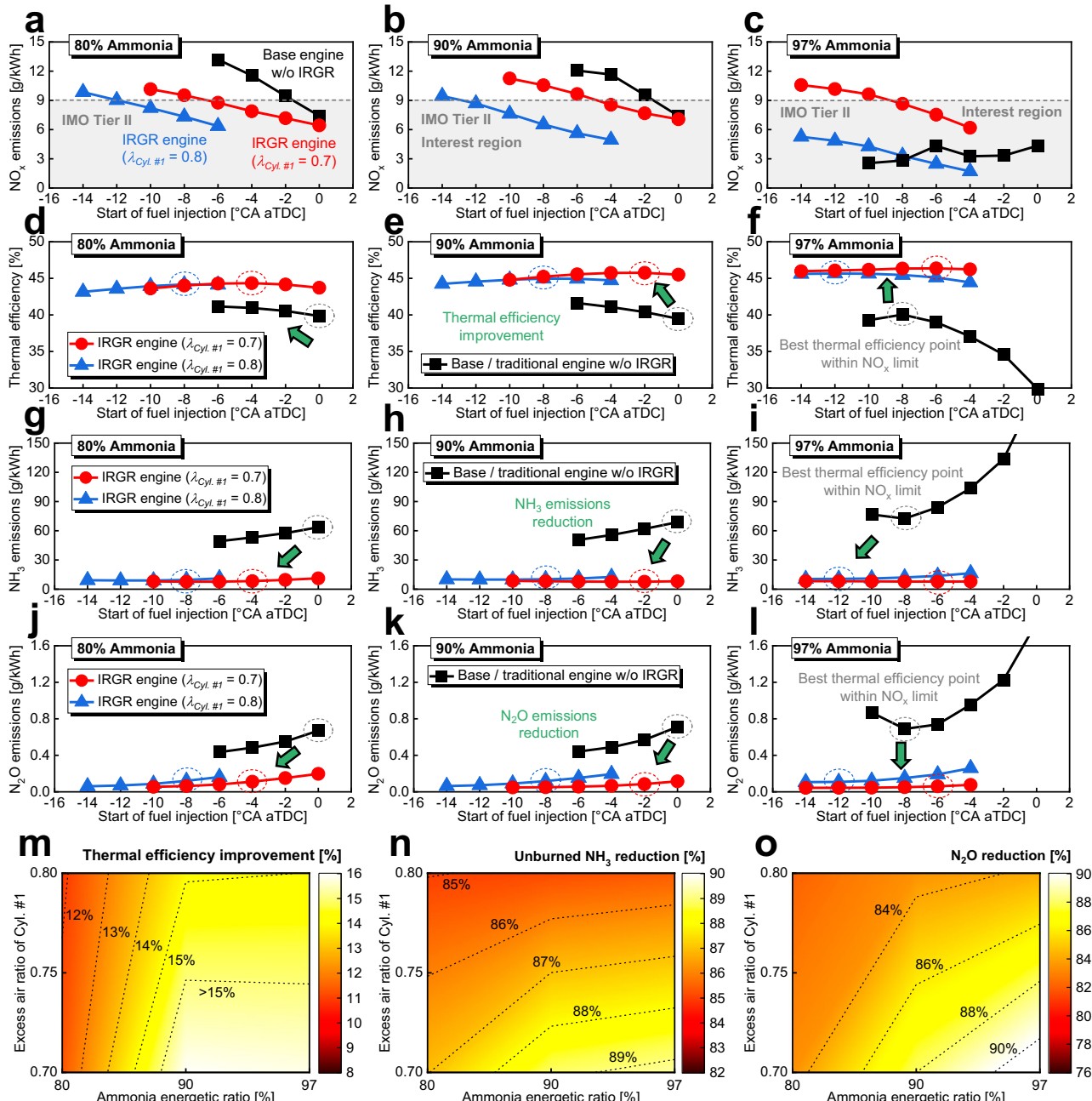

**Fig. 4 | Effects of IRGR technology under ammonia energetic ratio of 80%, 90%, and 97%. a–c** Comparison of $NO_x$ emissions, **d–f** Comparison of indicated thermal efficiency, **g–i** Comparison of unburned $NH_3$ emissions, **j–l** Comparison of $N_2O$ emissions between the base/traditional ammonia engine without IRGR system and IRGR ammonia engine under ammonia energetic ratio of 80%, 90% and 97%, **m** Effects of IRGR technology on indicated thermal efficiency improvement, **n** Unburned $NH_3$ emission reduction, **o** $N_2O$ emissions reduction. Here, $\lambda$ refers to the overall excess air ratio, and the object of comparison is the entire engine including all four cylinders. The best ITE points within the IMO Tier II $NO_x$ limit are highlighted by the dashed circles in **d–l**, and these best points are compared in **m–o**. (The following remain consistent for all the engine types and cylinders: 1000 rpm and intake temperature of 318 K; For the base/traditional ammonia engine and Cylinders #2–4 of the IRGR ammonia engine, the total input energy per cylinder is constant at 1717 J). Source data are provided as a Source Data file.

hydrogen is recirculated to Cylinders #2–4 as the fuel enrichment level of Cylinder #1 increases, which contributes to the reduction of unburned ammonia emissions, as shown in Fig. 4n. Moreover, as shown in Fig. 4o, the higher the fuel enrichment level in Cylinder #1, the more significant the $N_2O$ reduction. This is because the higher hydrogen addition in Cylinders #2–4 will increase the in-cylinder temperature and promote the $N_2O$ thermal decomposition reactions. Figure 4m shows that the IRGR technology is helpful for improving the indicated thermal efficiency, and this is mainly attributed to the improvement of combustion efficiency and reduction of unburned

ammonia by hydrogen addition. There is a general trend that the advantage of IRGR technology on thermal efficiency improvement and emission reduction is more pronounced at higher ammonia energetic ratios, indicating the potential of IRGR technology for increasing the diesel substitution ratio. Generally, when the ammonia energetic ratio varies from 80% to 97% and the overall excess air ratio of the dedicated reforming cylinder varies from 0.7 to 0.8, the IRGR concept can reduce the unburned $NH_3$ and $N_2O$ emissions by at least 85.0% and 82.1%, respectively, and increase the indicated thermal efficiency by at least 10.9%, demonstrating the effectiveness of IRGR technology. Especially,

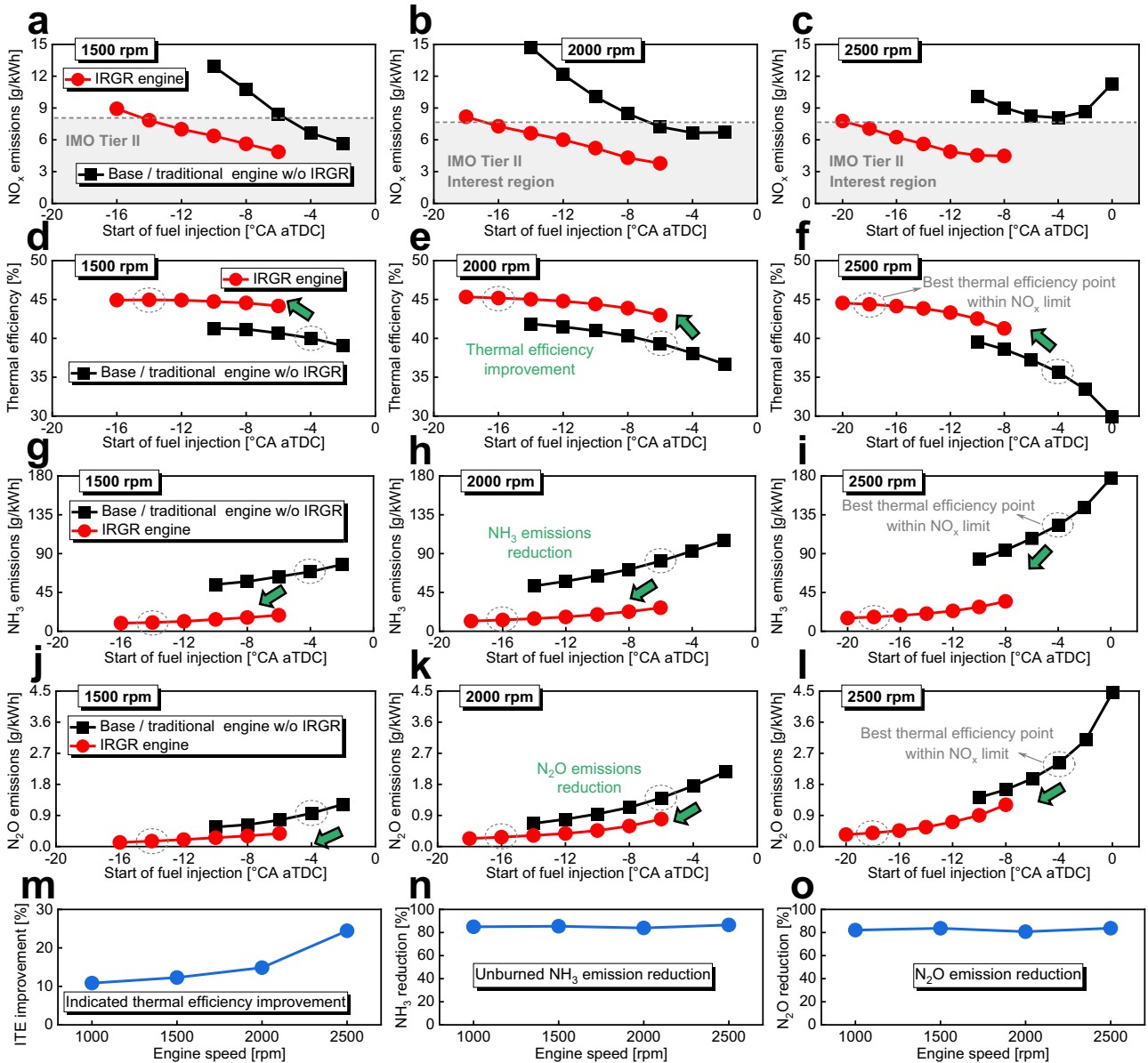

**Fig. 5 | Effects of IRGR technology under engine speeds of 1500, 2000, and 2500 rpm. a–c** Comparison of NO$_x$ emissions, **d–f** Comparison of indicated thermal efficiency, **g–i** Comparison of unburned NH$_3$ emissions, **j–l** Comparison of N$_2$O emissions between the base/traditional ammonia engine without IRGR system and IRGR ammonia engine under engine speed of 1500, 2000 and 2500 rpm, **m** Effects of IRGR technology on indicated thermal efficiency improvement, **n** Unburned NH$_3$ emission reduction, **o** N$_2$O emissions reduction. Here, the object of comparison is the entire engine including all four cylinders. For the IRGR ammonia engine, the

overall excess air ratio of the dedicated reforming cylinder is 0.8. The best ITE points within the IMO Tier II NO$_x$ limit are highlighted by the dashed circles in **d–l**, and these best points are compared in **m–o**. (The followings remain consistent for all the engine types and cylinders: ammonia energetic ratio of 80% and intake temperature of 318 K; For the base/traditional ammonia engine and Cylinders #2–4 of the IRGR ammonia engine, the total input energy per cylinder is constant at 1717 J). Source data are provided as a Source Data file.

at 97% ammonia energetic ratio and 0.7 overall excess air ratio of Cylinder #1, the IRGR engine can increase the indicated thermal efficiency by 15.8% and reduce the unburned NH$_3$ by 89.3%, N$_2$O by 91.2% compared to the base/traditional ammonia engine without IRGR.

Figure 5 shows the effectiveness of the IRGR concept under a higher engine speed (i.e., 1500, 2000, and 2500 rpm). Here, the ammonia energetic ratio is constant at 80%. Generally, increasing the engine speed will significantly reduce the indicated thermal efficiency and increase the unburned ammonia and N$_2$O emissions of base/traditional ammonia engines without IRGR. This is because the timescale for chemical reactions significantly decreases at higher engine speeds, thereby exacerbating the drawbacks of poor combustion characteristics of ammonia fuel. At 2500 rpm, the N$_2$O emissions of the base/

traditional ammonia engine without an IRGR system have exceeded 2.4 g/kWh in the interest region, leading to an equivalent GHG effect that is similar to that of the pure diesel mode. Undoubtedly, this will significantly hinder the further applications of traditional ammonia engines. Similar to Fig. 4, the best indicated thermal efficiency points within the IMO Tier II NO$_x$ limit are also highlighted by the dashed circles, and these best points are compared in Fig. 5m–o. Not surprisingly, the potential of IRGR technology to improve indicated thermal efficiency is more pronounced at higher engine speed as shown in Fig. 5m. Especially, at 2500 rpm, the IRGR concept can increase the indicated thermal efficiency by 24.4% compared to the base/traditional ammonia engine without IRGR. At the same time, it can reduce the unburned NH$_3$ by 86.6% and N$_2$O by 83.7%.

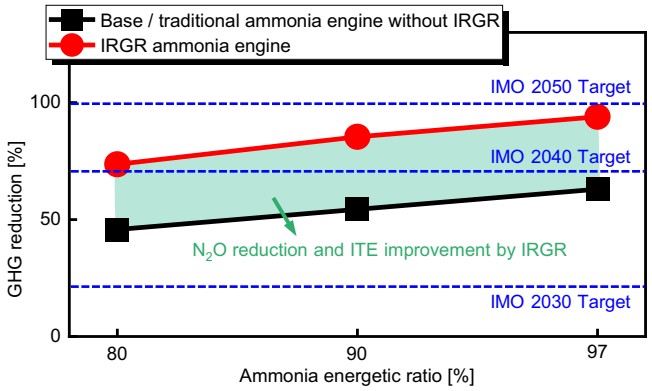

**Fig. 6 | Effects of IRGR technology on the reduction of greenhouse gas emissions under various ammonia energetic ratios.** Here, the data refers to the best indicated thermal efficiency points in Fig. 4, and the baseline is the traditional pure diesel mode. The object of comparison is the entire engine including all four cylinders, and the GHG emissions are calculated as $298 \times N_2O + CO_2$ (The following remain consistent for all the engine types and cylinders: 1000 rpm and intake temperature of 318 K; For traditional pure diesel engine, base/traditional ammonia engine, and cylinders #2–4 of the IRGR ammonia engine, the total input energy per cylinder is constant at 1717 J). Source data are provided as a Source Data file.

## IRGR technology for GHG reduction

Figure 6 shows the effects of IRGR technology on the reduction of GHG emissions (i.e., $298 \times N_2O + CO_2$) under various diesel energetic ratios. Here, the data refers to the best indicated thermal efficiency points in Fig. 4, and the baseline is the traditional pure diesel mode. For the base/traditional ammonia engine without an IRGR system, although 97% of the total energy is supplied by ammonia fuel, the IMO 2040 target of 70% GHG reduction remains unachievable. This is mainly because the formation of $N_2O$ emission, which has a greenhouse effect of around 298 times that of $CO_2$, offsets the benefits of GHG reduction. However, the IRGR ammonia engine can significantly reduce the GHG emissions compared to the base/traditional ammonia engine without the IRGR system since it can significantly reduce the $N_2O$ emissions and improve the thermal efficiency drastically. Especially, at a 3% diesel energetic ratio, the IRGR ammonia engine is able to reduce GHG emissions by 83.7% compared to the base ammonia engine without IRGR. At the same time, it is able to reduce GHG emissions by 94.0% compared to the traditional pure diesel mode.

## Discussion

In this paper, a concept termed in-cylinder reforming gas recirculation (IRGR) is initiated to simultaneously improve the thermal efficiency and reduce the unburned $NH_3$, $NO_x$, $N_2O$, and GHG emissions of pilot-diesel-ignition ammonia combustion engine. A chemical reaction mechanism for the combined ammonia and n-heptane combustion consisting of 65 species and 344 reactions (i.e., Supplementary Code 1 and Supplementary Code 2) is developed and integrated into the numerical simulation platform, which is then validated against the experimental data of a four-cylinder pilot-diesel-ignition ammonia engine without IRGR system. The potential of the IRGR concept on thermal efficiency improvement and emission reduction is numerically studied under various diesel energetic ratios, engine speeds, fuel injection timings, and fuel enrichment levels of Cylinder #1.

The simulations with the mechanism developed in this study can well predict the experimental heat release and in-cylinder pressure evolutions under various ammonia energetic ratios (40%, 60%, 80%, 90%) and engine loads (i.e., 50%, 75%, 85%, 100%), and exhibit the best predictions for exhaust emissions. For the dedicated reforming cylinder of the IRGR engine, the hydrogen is mainly generated near the high-temperature region, where the reactions of hydrogen production by ammonia reforming and decomposition are most positive. When the dedicated reforming cylinder operates at the fuel-rich condition of 0.7 and 0.8 overall excess air ratios, the hydrogen conversion ratios are 32.0% and 20.8%, respectively. Owing to the lack of oxygen, the formation of NO and $N_2O$ in the dedicated reforming cylinder is negligibly small compared to the base/traditional ammonia engine without an IRGR system. For the hydrogen-enriched cylinders (i.e., Cylinders #2–4) of the IRGR engine, the hydrogen addition increases the chemical reaction activity of the combustible mixture and reduces the quenching distance, which is helpful for significantly reducing the unburned ammonia in the clearance volume. Moreover, owing to the reduced oxygen content caused by reforming gas recirculation, the NO and $N_2O$ production reactions are also suppressed. At 3% diesel energetic ratio and 1000 rpm, the IRGR engine can increase the indicated thermal efficiency by 15.8% and reduce the unburned $NH_3$ by 89.3%, $N_2O$ by 91.2%, GHG (i.e., $298 \times N_2O + CO_2$) by 83.7% compared to the base/traditional ammonia engine without IRGR. At the same time, it is able to reduce carbon footprint by 97.0% and GHG by 94.0% compared to the traditional pure diesel mode. At 20% diesel and 2500 rpm, the IRGR concept can increase the indicated thermal efficiency by 24.4% and reduce the unburned $NH_3$ by 86.6% and $N_2O$ by 83.7% compared to the base/traditional ammonia engine without IRGR. The detailed 3D-CFD simulations suggest that the IRGR concept could solve the bottleneck problems such as the high $N_2O$ and unburned ammonia emissions, low thermal efficiency, and trade-off relationship between unburned $NH_3$ and $NO_x$ emissions in combustion optimization of traditional pilot-diesel-ignition ammonia combustion engine.

## Methods
### Experimental setups

In this paper, a four-cylinder ammonia-diesel dual-fuel engine with a 95 mm bore diameter without an IRGR system (i.e., the base engine) was used to acquire the experimental data for the validation of the chemical kinetic mechanism and numerical model. Figure 7a shows the schematic diagram of the whole system of the base engine, which mainly consists of the engine and dynamometer, ammonia fuel supply system, high-pressure pilot fuel injection system, intake system, and exhaust emission measurement system, while Fig. 7b displays a physical image of the base engine. Table 1 shows the engine specifications and experimental operating conditions under various ammonia energetic ratios. For the experiments, the gaseous ammonia was supplied at a pressure of 0.6 MPa before the intercooler, and the diesel fuel was directly injected into the cylinder at a pressure of 120 MPa. Here, the Brooks ammonia mass flow controller and AVL 735 s fuel mass flow meter were employed to obtain the mass flow rates of ammonia and diesel fuel, respectively. The HORIBA LI250 AC electrical dynamometer was utilized for brake torque measurement and engine speed control, NI PXI for the control of the engine system, and AVL GH15DK pressure transducers for the in-cylinder pressure measurement. Moreover, the exhaust $NO_x$ and $CO_2$ were measured by HORIBA MEXA-ONE-RS, and the exhaust unburned $NH_3$ and $N_2O$ were obtained by the HORIBA FTX-ONE-CS Fourier-transform infrared gas analyzer.

### Construction of numerical models

The numerical simulations were conducted utilizing the CONVERGE code package, and the engine combustion chamber geometries were consistent with the experimental engine in the section of 'Experimental setups'. The calculated window spanned from the intake valve closure to the exhaust valve open, including the compression, pilot-diesel injection, mixing of diesel spray and ammonia-air mixture, combustion, expansion, and heat transfer processes, and the in-cylinder working mediums were set to be uniformly mixed at the intake valve closure timing. For the pilot-diesel-ignition ammonia engine with IRGR system (i.e., IRGR engine), the simulations of the dedicated reforming cylinder under fuel-rich conditions (i.e., Cylinder

**Table 1 | Engine specifications and experimental operating conditions under various ammonia energetic ratios of the base engine without IRGR system for validating the chemical kinetic mechanism and numerical models**

| Parameters | Value | | | |
|---|---|---|---|---|
| Engine specification | 4 cylinders without IRGR | | | |
| Bore × stroke (mm) | 95×102 | | | |
| Nozzle hole diameter (mm) × number | 0.127×8 | | | |
| Compression ratio | ~17.5 | | | |
| Engine load (%) | 75 | | | |
| Brake mean effective pressure (MPa) | 0.7 | | | |
| Diesel injection pressure (MPa) | 120 | | | |
| Diesel injection timing (°CA aTDC) | −2 | −3 | −6 | −5 |
| Ammonia energetic ratio (%) | 40 | 60 | 80 | 90 |
| Ammonia supply pressure (MPa) | 0.6 | | | |
| Intake temperature (K) | 318 ± 3 | | | |
| Engine speed (rpm) | 1000 | | | |

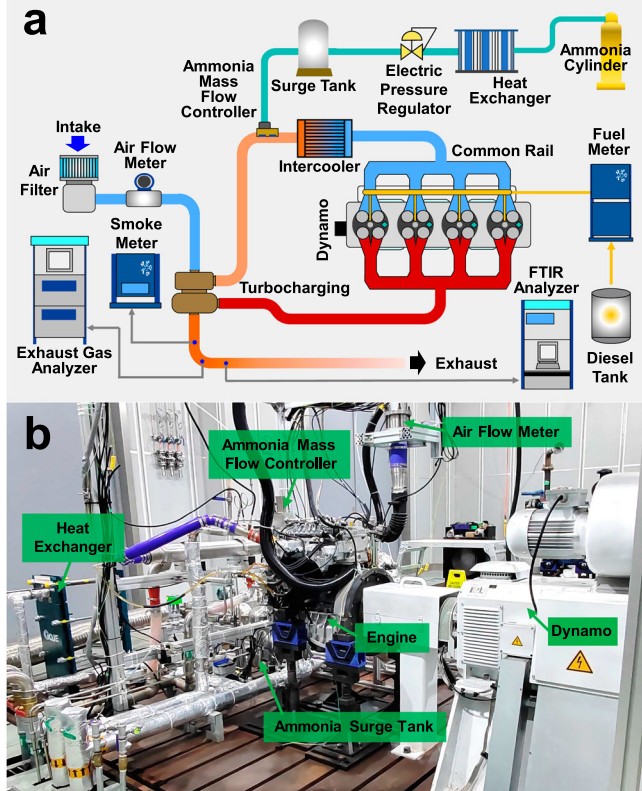

**Fig. 7 | Experiment setup for the pilot-diesel-ignition ammonia dual-fuel combustion engine without in-cylinder reforming gas recirculation (IRGR) system (i.e., the base engine). a** Schematic diagram of the whole engine system, **b** Engine physical image.

#1) were conducted first, and then the intake composition of the other three hydrogen-rich cylinders (i.e., Cylinders #2–4) was calculated according to the exhaust composition of the dedicated reforming cylinder. Figure 8a exhibits the details of the computational domain and mesh near the TDC. The base grid was set to 1.8 mm, the fixed mesh refinement was applied for the near-nozzle region at two levels, and the adaptive mesh refinement was used in the cylinder at two levels. For all the engine types and cylinders, the RANS equation coupled with RNG k-ε model[48] and Han and Reitz model[49] were employed for calculating the turbulence flow and heat transfer, respectively. The Lagrangian particle was employed to simulate the diesel fuel injection,

with the Kelvin-Helmholtz-Rayleigh-Taylor[50] for spray break-up, dynamic drag[51] for droplet drag, no-time-counter[52] for droplet collision, and Frossling[53] for droplet evaporation. The combustion processes were simulated utilizing the SAGE solver, and the latest ammonia/n-heptane chemical kinetic mechanism reported in 2023[54] (i.e., No. 1 in Table 2) and the mechanisms merged/developed by the present study (i.e., No. 2-No. 7 in Table 2) are compared with the experiment data in the following 'Validation of numerical models' section.

For the merged mechanisms (i.e., No. 2-No. 6), the n-heptane (i.e., C7) was selected as the surrogate for diesel fuel and merged with the ammonia chemical kinetic mechanism widely used in recent years[55–59]. The mechanism developed in this study (i.e., No. 7), has the same 344 reactions as No. 6, but the reaction constants of $NH_3 + OH = NH_2 + H_2O$, $NH_2 + NO_2 = H_2NO + NO$ and $NH_2 + NO_2 = N_2O + H_2O$ have been updated based on the latest literature. Specifically, the reaction constants for the first two reactions were obtained from[28], while the reaction constant for the third reaction was obtained from[58]. The reason for the above modification is that mechanism No. 6 performed relatively well among the first six mechanisms, but it still can not predict the experimental data of combustion phasing and unburned ammonia when the ammonia energetic ratio is larger than 60%, as shown in Fig. 8b–i. As a result, we updated the above three elementary reactions closely related to ammonia oxidation based on the more recent literature to satisfy the calibrating processes. The mechanism and thermal files of the mechanism developed in this study (i.e., No. 7) are given as Supplementary Code 1 and Supplementary Code 2, respectively.

### Validation of numerical models

In this section, the four-cylinder pilot-diesel-ignition ammonia combustion engine without IRGR system (i.e., the base engine) described in the section of 'Experimental setups' was used to obtain the experimental data for selecting the most suitable kinetic mechanisms and validating the present numerical models. Figure 8b–e shows the comparison between the experimental and simulated results for the apparent heat release rate and in-cylinder pressure of the base engine under ammonia energetic ratios of 40%, 60%, 80%, and 90%. Figure 8f–i shows the comparison of unburned $NH_3$, $NO_x$, $N_2O$, and GHG (i.e., $298 × N_2O + CO_2$) emissions of the base engine between the experiments and simulations using the chemical kinetic mechanisms listed in Table 2. Here, the engine load is 75%, engine speed is 1000 rpm, diesel fuel injection pressure is 120 MPa, intake temperature is 318 K, and more details are given in Table 1. It can be observed that the overall heat release and in-cylinder pressure evolutions by simulations with the mechanism developed in this study (i.e., No. 7) are in excellent agreement with the experimental data under ammonia energetic ratios of 40%, 60%, 80%, and 90%, while the simulations with the other mechanisms are almost unable to predict the ignition delay and combustion phasing as well as pressure evolution when ammonia energetic ratio is higher than 60%. The unburned $NH_3$ at a higher ammonia energetic ratio predicted by the simulations with mechanisms No. 1–6 is unacceptable, however, the simulations with the mechanism developed in this study (i.e., No. 7) exhibit accurate exhaust emission predictions. For instance, the prediction errors for the unburned $NH_3$ emissions are 7.8%, 5.6%, 3.5%, and 0.9% at ammonia energetic ratios of 40%, 60%, 80%, and 90%, respectively (i.e., Fig. 8f). Generally, the present numerical models with the mechanism developed in this study (i.e., No. 7) are reliable and accurate for predicting the ignition delay, heat release rate, in-cylinder pressure and main exhaust emissions of the ammonia-diesel dual-fuel engine under a wide range of ammonia energetic ratio, and these models are further used for evaluating the effectiveness of the IRGR concept.

The high precision and robustness of the 3D-CFD simulations are also validated against the experimental data under various engine

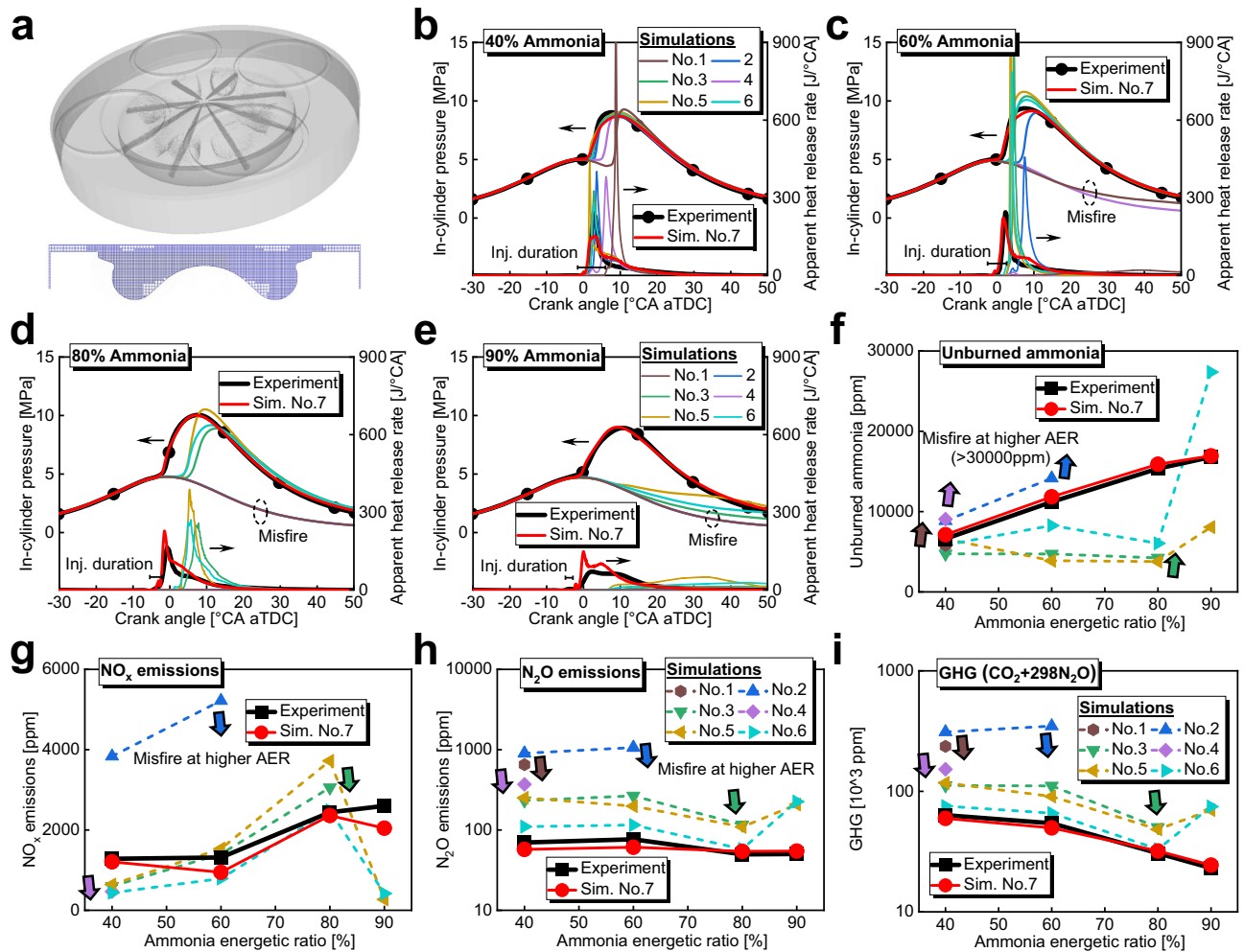

**Fig. 8 | Numerical setups and validation of the numerical models and chemical kinetic mechanisms. a** Details of the computational domain and mesh near the TDC, **b**–**e** Comparison of apparent heat release rate and in-cylinder pressure of the base engine between the experiments and simulations using the ammonia/n-heptane chemical kinetic mechanisms listed in Table 2 under ammonia energetic ratio of 40%, 60%, 80%, and 90%, **f** Comparison of unburned NH₃, **g** Comparison of NOₓ, **h** Comparison of N₂O, **i** Comparison of GHG (i.e., 298 × N₂O + CO₂) emissions of the base engine between the experiments and simulations using the ammonia/n-heptane chemical kinetic mechanisms listed in Table 2. (75% engine load, 1000 rpm, 120 MPa diesel injection pressure, and 318 K intake temperature). Source data are provided as a Source Data file.

**Table 2 | Existing and merged/developed chemical kinetic mechanism for ammonia/n-heptane combustion in this study**

| No. | Details | Type | Element | Species | Reaction |
|---|---|---|---|---|---|
| 1 | Xu (2023)[54] | Existing | 6 | 69 | 389 |
| 2 | GRI 3.0 (1999)[55] + C7 | Merged in this study | 5 | 70 | 420 |
| 3 | Mei (2019)[56] + C7 | Merged in this study | 6 | 65 | 406 |
| 4 | Okafor (2019)[57] + C7 | Merged in this study | 5 | 60 | 255 |
| 5 | Shrestha (2021)[58] + C7 | Merged in this study | 6 | 140 | 1163 |
| 6 | Song (2016)[59] + C7 | Merged in this study | 6 | 65 | 344 |
| 7 | Developed from No. 6 | Developed in this study | 6 | 65 | 344 |

loads (i.e., 50%, 75%, 85%, 100%), as shown in Supplementary Table 1 and Supplementary Fig. 1. Here, the followings are kept constant for all the cases: 80% ammonia energetic ratio, 1000 rpm engine speed, 120 MPa diesel injection pressure, and 318 K intake temperature. It can be observed that the overall heat release and in-cylinder pressure evolutions by simulations with the mechanism developed in this study (i.e., No. 7 and red line) are almost overlapped with the experimental data under all the engine loads, indicating the high precision and

robustness of the present 3D-CFD numerical simulations for the prediction of in-cylinder combustion processes. Meanwhile, at 80% ammonia energetic ratio here, the simulations with the other existing mechanisms are almost unable to predict the ignition timing and in-cylinder combustion phasing as well as pressure evolution for all the engine loads. It should be noted that in order to ensure the robustness and reasonableness for the subsequent use in IRGR study, numerical sub-models such as turbulence flow, heat transfer, spray and

combustion, and their relevant model constants are rigorously kept constant for all the above 49 validation cases. Since all the important settings from the experimentally verified 3D-CFD simulations such as engine specifications (i.e., bore, stroke, nozzle hole diameter, combustion chamber shape, and so on of the 95 mm base engine), sub-models and their relevant model constants, mesh methods and detailed chemical reaction mechanism (i.e., No. 7) are rigorously continued to be used for studying the IRGR concept, it is believed that the relevant calculated results are accurate and robust enough to support the highly exciting improvements by IRGR concept. Moreover, the detailed chemical reaction mechanism and thermal files developed in this study are added as Supplementary Code 1 and Supplementary Code 2 for further validation and use by academic and industrial sectors.

### Operating condition

Supplementary Table 2 summarizes the operating conditions for comparison of the base engine without the IRGR system and IRGR engine under various diesel energetic ratios (i.e., 3%, 10%, 20%). It is well-known that the diesel injection timing will influence the mixing of diesel spray and ammonia-air mixture, thereby affecting the in-cylinder working processes[60]. For a fair comparison, the pilot-diesel injection timing is studied within −20 to 0°CA aTDC for both the base engine and hydrogen-rich cylinders (i.e., Cylinders #2–4) of the IRGR engine to determine the optimal pilot-fuel injection timing, and then the engine performance and emissions of the IRGR engine and base engine are compared at this injection timing while keeping other operating conditions constant. It should be noted that to fully explore the potential of the IRGR concept, it is necessary to optimize the pilot-diesel injection timing for Cylinders #2–4 and the dedicated reforming cylinder (i.e., Cylinder #1) simultaneously. However, this will result in an unacceptable computational burden. As a result, for all the cases in the present study, the pilot-diesel injection timing of the dedicated reforming cylinder of the IRGR engine is not optimized and directly set to −6°CA aTDC, which is also helpful for highlighting the effectiveness of the IRGR concept. The total input energy per cylinder remains constant at 1717 J for the base engine and Cylinders #2–4 of the IRGR engine, while the value for the dedicated reforming cylinder of the IRGR engine is slightly higher, around 1988 J. Here, for the IRGR engine, the unburned fuel energy taken away by the exhaust gas from Cylinder #1 used for recirculation is excluded when calculating the input energy for Cylinder #1, instead, it is taken into account for the hydrogen-rich cylinders (i.e., Cylinders #2–4). Since the total input energy of the dedicated reforming cylinder is higher than that of the other cylinders, the IMEP of the dedicated reforming cylinder will be slightly higher, but all the cylinders of the IRGR engine maintain the IMEP of around 11.4 (±10%) bar at the best thermal efficiency point. For the real operation of an IRGR ammonia combustion engine, while the above IMEP difference is acceptable in terms of engine balance and vibration, it should be minimized as much as possible. For example, the IMEP difference between the dedicated reforming cylinder and Cylinders #2–4 can be alleviated by methods such as controlling the diesel injection timing and injection mass. However, as mentioned above, the control parameters such as diesel injection timing of the dedicated reforming cylinder are not optimized and directly set at −6°CA aTDC to avoid the unacceptable computational burden and highlight the effectiveness of the IRGR concept. Moreover, for a fair comparison, the diesel injection pressure (i.e., 120 MPa), engine speed (i.e., 1000 rpm), and intake temperature (i.e., 318 K) are controlled consistently for all the engine types and cylinders. The overall excess air ratio of the base engine without the IRGR system is 1.5, while under the influence of reforming gas recirculation, the value for Cylinders #2–4 of the IRGR engine is reduced to 1.1. It should be noted that the selected overall excess air ratio of the base engine without IRGR is relatively low compared to that of a general diesel engine, and this is because of the poor combustion characteristics of ammonia fuel and the aim of fair comparison with the IRGR engine.

In Supplementary Table 2, the overall excess air ratio of the dedicated reforming cylinder is 0.7. To evaluate the effects of fuel enrichment level in the dedicated reforming cylinder, the combustion and emission characteristics of the IRGR engine are also investigated when Cylinder #1 operates under a reduced fuel enrichment level of 0.8 overall excess air ratio. The main information for the 0.8 overall excess air ratio cases is consistent with Supplementary Table 2 and no details here. In the present study, the potential of IRGR technology is also evaluated at higher engine speeds of 1500, 2000, and 2500 rpm. For a fair comparison, under different engine speeds, the total input energy per cylinder is kept at 1717 J for the base engine and Cylinders #2–4 of the IRGR engine. Moreover, the diesel energetic ratio remains constant at 20%, diesel injection pressure is maintained at 120 MPa and intake temperature is kept at 318 K for all the engine types and cylinders.

## Data availability
All data within the manuscript and the supplementary files are available. Source data are provided in this paper.

## Code availability
The numerical simulations in this study were conducted utilizing the CONVERGE code package. A chemical reaction mechanism for the combined ammonia and n-heptane combustion consisting of 65 species and 344 reactions developed in this study is used for simulation, and the corresponding mechanism and thermal files are given in Supplementary Code 1 and Supplementary Code 2, respectively. The details for mesh setting and sub-models such as turbulence, heat transfer, fuel injection, spray break-up, dynamic drag, droplet evaporation, and combustion are given in the section of 'Methods'.

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

## Acknowledgements
This work was supported by the funding from National Natural Science Foundation of China (52301376 to X.Y.Z.), the Major International (Regional) Joint Research Project of National Natural Science Foundation of China (52020105009, T.L.), National Natural Science Foundation of China (52271325 to T.L.), Shanghai Rising-Star Program (Sailing Special Fund, 23YF1419700 to X.Y.Z.), China Postdoctoral Science Foundation (2022TQ0204 and 2022M722054 to X.Y.Z.), the financial support from Shanghai Jiao Tong University (X.Y.Z.), Dean's Chair Fund at the National University of Singapore (WBS No. E-465-00-0010-02 to W.M.Y.).

## Author contributions
X.Y.Z. conceived the concept, developed the chemical kinetic mechanism, conducted the numerical simulations, and wrote the paper; T.L. provided the experimental and numerical resources; X.Y.Z., R.C., Y.J.W., X.R.W., N.W., and S.Y.L. conducted the experiments; X.Y.Z., T.L., and W.M.Y. designed the research; T.L., M.K., and W.M.Y. contributed to writing and editing; T.L. and W.M.Y. supervised the study.

## Competing interests
The authors declare no competing interests.
