## [Peer Review File · Nature Communications]

Ammonia marine engine design for enhanced efficiency and reduced greenhouse gas emissionsReviewers' comments:

Reviewer #1 (Remarks to the Author):

This is an interesting idea but there is no experimental confirmation that operation with the IRGR system acts as promised. Therefore, it remains an interesting but unproven idea.

Specific comments:

- Were heat losses from the reforming cylinder considered?
- Line 168: Was heat loss considered as the exhaust from the reforming cylinder travels to the other cylinders?
- Line 92: Not sure what is meant by this "since partial of the high-reactivity diesel fuel"
- Line 181: Diesel contains aromatics but the surrogate (heptane) does not. What would be the effect?
- Line 287: "The interested overall excess air ratios" what do you mean by "interested"
- Line 362: Why do you say "could be" in "reasons could be the reduced oxygen content"? This is a model so you should be able to state why the model gave you these results.
- Line 516: This is just a model so you can not say "the IRGR concept is effective to solve the bottleneck problem". You can only say that "the model suggests that IRGR concept could solve the bottleneck problem"

Reviewer #2 (Remarks to the Author):

The topic of this paper seems very meaningful and interesting, but the depth and value of the research in the paper are still far from enough. Many questions are not clear, and there are no clear answers. Overall, it is not worth publishing in this journal. The specific comments are as follows:

- 1.The research topic of the paper is not very clear, what is near-zero emissions? This paper only proposes a low emission marine engine? Many issues are not specific and clear, and are too vague.
- 2.Is the research proposed in the paper on greenhouse gas emissions? However, from the specific content of the paper, the issue of other harmful gas emissions has been studied and analyzed, which makes readers confused. Is the author's research focus on greenhouse gases or various gas emissions?
- 3.The paper does not have a clear research conclusion on which fuel to use in the future maritime sector, and the author did not provide useful conclusions through comparative research on different fuels, which is the biggest deficiency of this paper.
- 4.The author completely neglected how to verify the emission reduction effect obtained in the paper.
- 5.The limitation of this paper lies in why the author only chose ammonia as a new marine fuel for research. Is there any evidence that the next generation of marine engines will necessarily burn ammonia?
- 6.From the current development status of international maritime industry, it is still necessary to demonstrate which type of fuel, such as methanol, ammonia, or hydrogen, will be used as the fuel for the next generation of ship engines. If the author can comprehensively compare and study these fuels from various aspects such as emissions, economy, and safety, and provide clear conclusions, then this paper will be very valuable. However, at present, this paper is relatively subjective and does not conduct comprehensive research and comparison on the technical performance and feasibility of different fuels, which makes its academic value far from sufficient.

Reviewer #3 (Remarks to the Author):

This article has a state-of-the-art results for ammonia-diesel multi cylinder engine operation with dedicated EGR concept. 3D CFD analysis was used after proper validation process of real engine operation. The results have a good quality and are timely appropriate on demand of decarbonization for maritime propulsion. I think that after the revision process corresponding to some details, this article can be accepted to this journal.

1. page 5, line 118

: As far as I know, partial oxidation of ammonia, under deficient amount of oxygen smaller than that of stoichiometry, has low yield of hydrogen compared to ammonia decomposition itself. The author should clarify the reason why partial oxidation was used rather than decomposition - one of the expected reasons is that to run the entire engine successfully, exothermic reaction should be required for every cylinder.

2. page 5, line 126

: I'm afraid but there is no citation for some literatures related to the concept of dedicated EGR which is similar to IRGR. Although the application of 'dedicated EGR' to ammonia-fueled engine has its novelty as first attempt, the author should clearly mention the history of dedicated EGR and what the difference is between that and IRGR in this study.

3. page 5, line 132

: It was mentioned that experiments were conducted without IRGR which was numerically considered. Thus, the author should mention some possible problems that might happen in real operation of IRGR such as unbalanced torque distribution and vibration, backfire of reformed hydrogen over certain fraction.

4. page 6, line 140

: I'm afraid but the readability of figure 2 is not well due to the large number of sub-figures. I would suggest to split figure 2 into three pieces - figure 2a, figure 2b, figure 2cdef, and figure 2ghi. If there is maximum number of figures in guideline, please ignore this comment.

5. page 8, Table 2

: Considering the mention related to the comparison between No.6 and 7 in Table 2, should the number of reaction in No.6 be 341, rather than 344?

6. page 9, line 208

: As far as I understand, newly-added three elementary reactions for No.7 in Table 2, as mentioned in the last part of section 2.2, have an important role to satisfy the matching process between numerical and experimental data. Please clarify the reason otherwise the author should mention that it is beyond the scope of this work.

7. page 10, line 235

: The original excess air ratio is relatively low compared to that of general CI diesel engine. Please clarify the reason why this value was selected - for example, low exhaust gas temperature and/or combustion instability due to ammonia.

8. page 11, line 257

: I'd like to suggest using mole fraction rather than mass fraction because 0.8 % of hydrogen mass fraction can be considerable as considering its extremely low molecular level.

Point-by-point response to the reviewers

NCOMMS-23-24260

Contents

1. To Reviewer #1: Point-by-point response	P2-5
2. To Reviewer #1 and #2: Validation of 3D-CFD simulations against more experiment data	P6-12
3. To Reviewer #2: Point-by-point response	P13-21
4. To Reviewer #2: Techno-economic study on various alternative marine fuels	P22-44
5. To Reviewer #3: Point-by-point response	P45-49

1. To Reviewer #1: Point-by-point response

This is an interesting idea but there is no experimental confirmation that operation with the IRGR system acts as promised. Therefore, it remains an interesting but unproven idea.

Re: Thank you very much for your kind review and confirming the interest of this study. We could not agree more that experimental validation is more convincing than 3D-CFD simulation validation. Unfortunately, very few institutions have the ability to conduct marine engine experiments owing to its large size and extremely high cost, especially for the ammonia-fueled marine engines that have just become a research hotspot in recent years. As a result, numerical simulation has been deemed as a necessary step at the early stage of any new engine development. Moreover, the IRGR ammonia engine concept was newly proposed by us, and there is no similar ammonia engine available worldwide for experimental validation. In fact, we have already initiated the design and strength check of the IRGR engine. However, due to the relatively weaker manufacturing and system integration ability of universities compared to industries, even a moderate modification to the engine like the IRGR technology will usually take us more than 2 years and cost a large amount of money. Based on our recent successful experience in designing two "traditional" ammonia combustion engines (i.e. **Fig. R2** and **Fig. R6** in section "*Validation of 3D-CFD simulations against more experiment data*" in Pages 6-12 of this response file, the "traditional" here is compared to the new IRGR engine, it is also a dual fuel engine), we are confident in the successful operation of the IRGR engine in the future, but we believe that our current highly exciting 3D-CFD simulation results based on extensive experimentally verified high-precision numerical simulation model and robust chemical reaction mechanism will provide significant and invaluable inspirations to the industries who are also eagerly dedicated to the development of next-generation marine engines, thus collectively contributing to the achievement of the very urgent IMO GHG target.

We are very excited that the "Computational and simulation studies" are in the scope of Applied Science and Engineering Research @ Nature Communications, and we agree that only experimentally validated high-precision and convincing numerical studies will be considered. As you know, the ammonia-fueled engines have just become a research hotspot in recent years, and all the existing numerical simulations and chemical kinetic mechanism can only predict the engine performance and emissions data with an up to 60% ammonia energetic ratio to some extent, which is far behind the high-precision prediction of 90% ammonia energetic ratio and above required by the present study. As a result, before the study of IRGR concept, we dedicated our most efforts and time in the past two years to establishing our 3D-CFD simulation models together with detailed chemical kinetic mechanism consisting of 65 species and 344 reactions, and verify them against our state-of-the-art ammonia engine experimental data with ammonia energetic ratios of up to 90%. For the 3D-CFD simulation models, the engine combustion chamber geometries were rigorously consistent with the

experiment engine, and all the piston compression, pilot-diesel injection, spray break-up, evaporation, mixing of diesel spray and ammonia-air mixture, combustion, expansion and in-cylinder heat transfer processes have been well considered. As shown in the following section "*Validation of 3D-CFD simulations against more experiment data*" (i.e. Pages 6-12 in this response file), a total of 56 validation cases have been established and verified against the experimental data, and the predicted ignition delay, combustion phasing, heat release rate, in-cylinder pressure evolution and pollutant emissions including unburned ammonia, N₂O, NO_x and GHG agree very well with the experimental data under a wide range of ammonia energetic ratios (i.e. 40%, 60%, 80%, 90%) and engine loads (newly added in this revision, i.e. 50%, 75%, 85%, 100%), indicating the high-precision and robustness of our 3D-CFD numerical simulation models and detailed chemical reaction mechanism. Since all the important settings from the above experimentally verified 3D-CFD simulations such as engine specifications (i.e. engine bore diameter, stroke, combustion chamber geometries, and so on of the base engine), sub-models (i.e. spray, combustion, heat transfer, turbulence, heat transfer, and so on) and their relevant model constants, mesh methods and chemical reaction mechanism are rigorously continued to be used for the study of IRGR concept, we believe that the relevant calculated results are robust enough to support the highly exciting improvements by IRGR concept.

Moreover, to the best of our knowledge, we strongly believe that this is the first numerical simulations that have undergone experimental validation for pilot-diesel-ignited ammonia engine operating at ammonia energetic ratio exceeding 80%, and the accuracy and reliability of our CFD simulations are significantly higher than the state-of-the-art ammonia simulation studies. The newly developed detailed chemical reaction mechanism and thermo files have been added as Appendix 1 and 2 for further validation and use by academic and industrial sectors. Please review section "*Validation of 3D-CFD simulations against more experiment data*" (i.e. Pages 6-12 in this response file) for more details, and we hope the above clarification and relevant experimental validations are helpful.

The responses to your other invaluable questions are added below:

1. Were heat losses from the reforming cylinder considered?

Re: The in-cylinder heat transfer losses from the reforming cylinder have been well considered and we have highlighted this information in line 320 of the revised manuscript as "*For all the engine types and cylinders, Amsden model [52] was employed for calculating heat transfer*". As shown in the **Fig. R3** and **Fig. R5** in section "*Validation of 3D-CFD simulations against more experiment data*" (i.e. Pages 6-12 in this response file), the calculated apparent heat release rate, which is the difference between the heat release rate and in-cylinder heat transfer rate, agree very well with the experimental data under a wide range of operating conditions, indicating the effectiveness and accuracy of the present detailed 3D-CFD simulations as well as the selected combustion and heat transfer models. As shown in **Fig. R1** here or Fig. 10b of the revised manuscript, we have also provided the data of in-

cylinder heat transfer losses (i.e. cooling losses) between the base/traditional ammonia combustion engine without IRGR system and IRGR ammonia combustion engine. Here, the object of the comparison is the entire engine including all the four cylinders (i.e. one reforming cylinder and three hydrogen-rich cylinders for IRGR engine). It can be seen that the in-cylinder heat transfer loss of the base engine is around 16.9%, while the IRGR engine has an increased heat transfer loss of around 21.0% due to increased temperature as a result of improved combustion. Excitingly, the IRGR engine can significantly reduce the unburned loss from 11.6% to around 3%. Since the reduction of unburned loss is larger than the increase of heat transfer loss, the IRGR concept can obviously increase the thermal efficiency.

Fig. R1. Comparison of heat balance between the base/traditional ammonia engine without IRGR system and IRGR ammonia engine. Here, the object of comparison is the entire engine including all the four cylinders. (The followings remain consistent for all the engine types and cylinders: ammonia energetic ratio of 90%, 1000 rpm, intake temperature of 318 K and diesel injection timing of -6°CA aTDC; For the base/traditional ammonia engine and Cylinders #2-4 of the IRGR ammonia engine, the total input energy per cylinder is constant at 1717 J).

2. Line 168: Was heat loss considered as the exhaust from the reforming cylinder travels to the other cylinders?

Re: As shown in **Fig. R1** here or Fig. 10b in the revised manuscript, the exhaust heat loss for both base engine and IRGR engine are similar (i.e. around 30%). In the present study, in order to highlight the effectiveness of IRGR concept and provide a fair comparison, we did not recover the exhaust gas heat loss from the reforming cylinder (i.e. heat loss considered as the exhaust from the reforming cylinder **not** travels to the other cylinders), and therefore the overall fuel economy can further increase if the exhaust heat loss recovery is considered. This information has provided in the last row of Table 3 (i.e. the intake temperature is kept at near room temperature 318 K for all the engine types and cylinders).

3. Line 92: Not sure what is meant by this “since partial of the high-reactivity diesel fuel”

Re: Sorry for the unclear statement. This statement is related to the split diesel injection strategy. For the split injection strategy, the pilot fuel (i.e. part of the injected diesel fuel) is injected first to mix with the ammonia-air mixture to improve the chemical reaction activity of the combustible mixture. We have rephrased this sentence in line 215 of the revised manuscript: *"With the split diesel injection strategies, since part of the high-reactivity diesel fuel (i.e. pilot fuel) has been fully mixed with ammonia-air mixture during the compression stroke..."*.

4. Line 181: Diesel contains aromatics but the surrogate (heptane) does not. What would be the effect?

Re: Owing to the similar chemical characteristics, heptane as surrogate for diesel has been widely used for diesel engine simulations, and most of the previous numerical studies have employed this approach to avoid the unacceptable computational burdens. We agree that the aromatics in diesel may affect the soot formation processes, but having negligible effect on ignition properties and combustion process. Moreover, the focus of the present study is ignition properties, heat release rate, in-cylinder pressure and engine-out emissions such as unburned ammonia, NO_x, N₂O and GHG rather than soot emissions, and the accuracy of simulation in predicting the above key parameters has been experimentally validated in the following section "*Validation of 3D-CFD simulations against more experiment data*" (i.e. Pages 6-12 in this response file). Furthermore, in the present study about IRGR pilot-diesel-ignition ammonia combustion engine, we focus on the ammonia energetic ratio between 80% and 97%. In another words, the diesel energetic ratio is as low as 3% to 20% (i.e., the diesel mass fraction is as low as 1.4% to 10%), and the influence of aromatics is almost negligible due to the extremely low percentage.

5. Line 287: “The interested overall excess air ratios” what do you mean by “interested”

Re: The "interested" represents what we are concerned about or the range of overall excess air ratios we studied. In line 427 of the revised manuscript, we have revised this statement to: *"the overall excess air ratios concerned here..."*.

6. Line 362: Why do you say “could be” in “reasons could be the reduced oxygen content”? This is a model so you should be able to state why the model gave you these results.

Re: We have corrected this typo and use *"can be attributed to"* in line 533 of the revised manuscript according to your kind reminder.

7. Line 516: This is just a model so you can not say “the IRGR concept is effective to solve the bottleneck problem”. You can only say that “the model suggests that IRGR concept could solve the bottleneck problem”

Re: Thanks a lot for your kind suggestion. We have used more precise expressions in line 695 of the revised manuscript. *"The detailed 3D-CFD simulations suggest that the IRGR concept could solve the bottleneck problem..."*.

2. To Reviewer #1 and #2: Validation of 3D-CFD simulations against more experiment data

Summary of the experimental validations

During the revision processes, we have validated our 3D-CFD numerical simulation models and detailed chemical reaction mechanism against more experiment data from our state-of-the-art pilot-diesel-ignition ammonia combustion engine without IRGR system to further demonstrate their high-precision and robustness. As shown in the following two sections of "*Experimental validations*", we have established a total of 56 validation cases under various ammonia energetic ratios (i.e. 40%, 60%, 80%, 90%, 93%), engine loads (i.e. 50%, 75%, 85%, 100%) and engine bore diameters (i.e. base engine with 95 mm bore diameter, as shown in **Fig. R2**, and a larger engine with 175 mm bore diameter that we recently finished, as shown in **Fig. R6**) to verify the reliability of our 3D-CFD simulations. To ensure the robustness and reasonableness for the subsequent use in IRGR study, numerical sub-models such as turbulence flow, heat transfer, spray and combustion and their relevant model constants are rigorously kept constant for all the 56 validation cases. As shown in the following two sections of "*Experimental validations*", the simulations with the newly developed chemical reaction mechanism (i.e. No. 7) can well predict the ignition delay, combustion phasing, heat release rate, in-cylinder pressure evolution, and emissions including unburned ammonia, N₂O, NO_x and GHG under a wide range of operating conditions, indicating the high-precision and robustness of our 3D-CFD numerical simulations and chemical reaction mechanism. Since all the important settings from the experimentally verified 3D-CFD simulations such as engine specifications (i.e. bore, stroke, nozzle hole diameter, combustion chamber shape, and so on of the 95 mm base engine), sub-models and their relevant model constants, mesh methods and detailed chemical reaction mechanism (i.e. No. 7) are rigorously continued to be used for studying the IRGR concept, we believe that the relevant calculated results are accurate and robust enough to support the highly exciting improvements by IRGR concept. We hope the experimental validations in this section will be helpful, and more details are given as follows:

Experimental validations using the base engine under various ammonia energetic ratios (i.e. 40%, 60%, 80%, 90%) and engine loads (i.e. 50%, 75%, 85%, 100%)

Fig. R2 shows the experiment setup for our state-of-the-art pilot-diesel-ignition ammonia combustion engine without IRGR system (i.e. the base/traditional engine). Here, **Fig. R2a** shows the schematic diagram of the whole engine system, while **Fig. R2b** is the engine physical image. The engine bore diameter of the base engine is 95 mm, and more details about the engine specifications and experimental setups can be found in Section 2.1 of the revised manuscript. It should be noted a total of 49 validation cases under various ammonia energetic ratios (i.e. 40%, 60%, 80%, 90%) and engine loads (i.e. 50%, 75%, 85%, 100%) as well as chemical reaction mechanisms are established based on this base engine, and our IRGR study in the present work is also based on this base engine.

Fig. R2. Experiment setup for the pilot-diesel-ignition ammonia dual fuel combustion engine without in-cylinder reforming gas recirculation (IRGR) system (i.e. the base engine). (a) schematic diagram of the whole engine system, (b) engine physical image.

As shown in **Fig. R3** and **Fig. R4**, we have specifically established 28 validation cases coupled with detailed chemical kinetic mechanism and compared them against experimental engine data of the base engine (i.e. **Fig. R2**) under a wide range of ammonia energetic ratios (i.e. 40%, 60%, 80% and 90%) to demonstrate the accuracy and reliability of our numerical models and chemical kinetic mechanism. Here, No. 7 is the new chemical kinetic mechanism developed by this study, while mechanisms No. 1 to No. 6 are merged based on the existing ammonia chemical kinetic mechanism widely used in recent years, and more details can be found in Table 2 of the revised manuscript. Moreover, the validation for GHG emissions (i.e. **Fig. R4d**) is newly added in this revision and has been supplemented to the revised manuscript. It can be observed that the overall heat release and in-cylinder pressure evolutions by simulations with the newly developed mechanism (i.e. No. 7) are almost overlapped with the experimental data under all the ammonia energetic ratios (i.e. **Fig. R3**),

while the simulations with the other existing mechanisms (i.e. No.1-6) are almost unable to predict the ignition delay and combustion phasing as well as pressure evolution when ammonia energetic ratio is higher than 60%. The prediction errors for the unburned ammonia emissions are 7.8%, 5.6%, 3.5% and 0.9% at ammonia energetic ratios of 40%, 60%, 80% and 90%, respectively (i.e. **Fig. R4a**), while the prediction errors for the GHG emissions are 5.9%, 8.5%, 4.4% and 5.8% at ammonia energetic ratios of 40%, 60%, 80% and 90%, respectively (i.e. **Fig. R4d**). To the best of our knowledge, we strongly believe that this is the first numerical simulations that have undergone experimental validation for pilot-diesel-ignited ammonia engine operating at ammonia energetic ratio exceeding 80%, and the accuracy and reliability of our CFD simulations are significantly higher than the state-of-the-art ammonia simulation studies.

Fig. R3. Comparison of apparent heat release rate and in-cylinder pressure of the base engine between the experiments and simulations using the ammonia/n-heptane chemical kinetic mechanisms listed in Table 2 of the revised manuscript. Ammonia energetic ratio of (a) 40%, (b) 60%, (c) 80%, (d) 90%. (75% engine load, 1000 rpm, 120 MPa diesel injection pressure and 318 K intake temperature).

Fig. R4. Comparison of (a) unburned NH₃, (b) NO_x, (c) N₂O and (d) GHG (i.e. 298 × N₂O + CO₂) emissions of the base engine between the experiments and simulations using the ammonia/n-heptane chemical kinetic mechanisms listed in Table 2 of the revised manuscript. (75% engine load, 1000 rpm, 120 MPa diesel injection pressure and 318 K intake temperature).

In order to further prove the high-precision and robustness of our 3D-CFD numerical simulation model and detailed chemical reaction mechanism, during the revision processes, we have further established 21 validation cases and compared them against experimental engine data of the base engine (i.e. **Fig. R2**) under a wide range of engine loads (i.e. 50%, 85%, 100%), as shown in **Fig. R5**. Here, the black line refers to the experimental data, while the others indicate the simulations with different chemical reaction mechanisms. The followings are kept constant for all the cases: 80% ammonia energetic ratio, 1000 rpm engine speed, 120 MPa diesel injection pressure and 318 K intake temperature. It can be observed that the overall heat release and in-cylinder pressure evolutions by simulations with the newly developed mechanism (i.e. No. 7 and red line) are almost overlapped with the experimental data under all the engine loads, indicating the high-precision and robustness of the present 3D-CFD numerical simulations for the prediction of in-cylinder combustion processes. At 80% ammonia energetic ratio here, the simulations with the other existing mechanisms are almost unable to predict the ignition timing and in-cylinder combustion phasing as well as pressure evolution for all the engine loads, which is consistent with the existing ammonia engine simulation studies to date. We

have added the above experimental validations under various engine loads into the Appendix 3 of the revised manuscript to avoid too long the paper. Moreover, the newly developed detailed chemical reaction mechanism and thermo files are also added as Appendix 1 and Appendix 2 for further validation and use by academic and industrial sectors.

Fig. R5. Comparison of apparent heat release rate and in-cylinder pressure of the base engine between the experiments and simulations using the ammonia/n-heptane chemical kinetic mechanisms listed in Table 2. Engine load of (a) 50%, (b) 75%, (c) 85%, (d) 100%. (80% ammonia energetic ratio, 1000 rpm, 120 MPa diesel injection pressure and 318 K intake temperature).

Experimental validations using a larger pilot-diesel-ignited ammonia marine engine with 175 mm bore diameter that we recently finished (For validation of 3D-CFD simulations only)

The above experimental validations under various ammonia energetic ratios and engine loads use the experimental data from the base ammonia engine with 95 mm bore diameter (i.e. Fig. R2). In July 2023, we successfully finished the establishment of another "traditional" pilot-diesel-ignited ammonia marine engine with a larger bore diameter of 175 mm (i.e. Fig. R6, the "traditional" here is compared to the new IRGR engine, it is also a dual fuel engine), and achieved the stable operation under 90%+ ammonia energetic ratio (Official News in July 2023: *China's first large-bore medium-speed pilot-diesel-ignited ammonia marine engine with 90%+ ammonia energetic ratio*, https://mp.weixin.qq.com/s/eg-1U_V3VzeTitbpJZLATw). Although the validation of the IRGR

concept in the present study is based on the base engine with 95 mm bore diameter in **Fig. R2** rather than this 175 mm engine in **Fig. R6**, during the revision processes, the experimental data obtained from this 175 mm ammonia marine engine is also used for further demonstrating the reliability of our numerical sub-models and detailed chemical reaction mechanism. For the 3D-CFD simulations of the 175 mm engine, the engine geometry specifications such as bore, stroke, nozzle hole diameter and combustion chamber shape were rigorously consistent with the experiment engine in **Fig. R6**, while the numerical sub-models such as turbulence, spray and combustion and their relevant model constants were rigorously consistent with the simulations of the 95 mm engine.

Fig. R6. Experiment setup for the pilot-diesel-ignition ammonia dual fuel combustion single cylinder engine with 175 mm bore diameter that we recently finished. (a) schematic diagram of the whole engine system, (b) engine physical image, (c) engine specifications and operating conditions. (For validation of 3D-CFD simulations only)

As shown in **Fig. R7**, under 93% ammonia energetic ratio, 1000 rpm engine speed, 15 bar indicated mean effective pressure (IMEP), 150 MPa pilot diesel injection pressure and 318 K intake temperature, the overall heat release and in-cylinder pressure evolutions by simulation with the newly developed mechanism (i.e. No. 7) agree very well with the experimental data, which once again demonstrates the high-precision and robustness of our simulation models and chemical reaction mechanism. It should be noted again that the above experimental validations based on this 175 mm engine primarily serve to demonstrate the reliability and robustness of our numerical sub-models and chemical reaction mechanism, and the study of IRGR concept, which is the focus of the present study, is not based on this engine but on the base engine with 95 mm bore diameter in **Fig. R2**. As a result, we have placed the above experimental validations about this 175 mm engine into the Appendix 3 of the revised manuscript to avoid too long the paper.

Fig. R7. Comparison of apparent heat release rate and in-cylinder pressure of the 175 mm ammonia combustion engine between the experiments and simulations using the ammonia/n-heptane chemical kinetic mechanisms listed in Table 2 of the revised manuscript. (93% ammonia energetic ratio, 1000 rpm, 15 bar IMEP, 150 MPa pilot diesel injection pressure and 318 K intake temperature).

3. To Reviewer #2: Point-by-point response

The topic of this paper seems very meaningful and interesting, but the depth and value of the research in the paper are still far from enough. Many questions are not clear, and there are no clear answers. Overall, it is not worth publishing in this journal. The specific comments are as follows:

Re: Thank you a lot for your kind review and confirming the high interest and meaningful of this work. We have tried our best to clarify the issues you kindly mentioned and answered all your questions. Especially, during the revision processes, we have dedicated our most efforts to conducting a detailed techno-economic study on various alternative marine fuels to further improve the depth and value of this work, as shown in section "*Techno-economic study on various alternative marine fuels*" of this response file (i.e. Pages 22-44) or Appendix 4 of the revised manuscript. In this techno-economic study, we conduct a comprehensive comparison of various alternative fuels (i.e. hydrogen, ammonia, methanol, methane) from five perspectives: economy, fuel availability, technology readiness level, safety and emissions, and clear conclusions have been drawn based on the main results. For the first time in the economic analysis, all the costs related to the use of alternative fuels have been considered as comprehensively as possible, including the fuel cost, on board fuel storage cost, engine and ship cost, extra voyage cost caused by the sacrifice of cargo capacity, GHG penalty by Well-to-Propeller CO₂ and CH₄ emissions, and GHG penalty by N₂O emissions from the current ammonia engine technology. More details about the Point-by-point responses and section "*Techno-economic study on various alternative marine fuels*" are given as follows:

1. The research topic of the paper is not very clear, what is near-zero emissions? This paper only proposes a low emission marine engine? Many issues are not specific and clear, and are too vague.

Re: Thank you for your kind reminder, we have added the specific definition of "near-zero emissions in maritime sector" at the beginning of the Introduction section in our revised manuscript, the relevant text is given here as follows:

"According to the latest definition in 2023 by Lloyd's Register (LR) [1], "zero greenhouse gas (GHG) emissions in maritime sector" can be divided into: (1) Absolute-zero emissions mean that there are no GHG emissions from the whole Well-to-Propeller (WtP) lifecycle; (2) Net-zero emissions can be achieved when anthropogenic GHG emissions to atmosphere can be balanced by anthropogenic removal from the perspective of WtP lifecycle; (3) Near-zero emissions refer to a reduction of more than 80% in GHG emissions compared to low-sulphur fuel oil (LSFO). On July 7, 2023, the International Maritime Organization (IMO) proposed a reduction of at least 70% in GHG from international ships by 2040, with the ultimate goal of achieving net-zero GHG shipping by around 2050 [2]. As a result, replacing the marine oil fueled diesel engines, which account for around 98.8% of the prime movers in international ships [3], with alternative fuel fueled engines that feature the

near-zero or net-zero GHG emissions is inevitable to achieve the IMO target in the next 30 years."

For the ammonia combustion engine, the N₂O emission that has a greenhouse effect around 298 times that of CO₂ must be considered for calculation of GHG emissions. In the present study, at 3% diesel energetic ratio, the IRGR engine shows the potential to reduce GHG emissions (i.e. $298 \times \text{N}_2\text{O} + \text{CO}_2$) by 94.0% compared to the traditional fuel oil fueled diesel engine, while the IRGR engine can reduce GHG by 83.7% compared to the state-of-the-art ammonia engine without IRGR system (i.e. base/traditional ammonia engine mentioned in the present study). As a result, the research topic of this paper belongs to the "near-zero" according to the definition of LR. Owing to the stability of the ultra-short pulse diesel injection, we did not further reduce the diesel fraction to explore the maximum reduction of GHG emissions. We believe that with the development of corresponding diesel injectors and the use of biodiesel as pilot fuel as well as the proposal of appropriate ignition strategies such as pre-chamber ignition system (i.e. without pilot-fuel), the IRGR concept has the potential to further reduce GHG emissions. It should be noted that for engine combustion, there is no difference among Gray, Blue and Green ammonia, and therefore the Green ammonia that is highly endorsed by maritime sector is considered here. More details about the definition and difference about Gray, Blue and Green fuels can be found in section "*Techno-economic study on various alternative marine fuels*" of this response file (i.e. Pages 22-44) or Appendix 4 of the revised manuscript.

2. Is the research proposed in the paper on greenhouse gas emissions? However, from the specific content of the paper, the issue of other harmful gas emissions has been studied and analyzed, which makes readers confused. Is the author's research focus on greenhouse gases or various gas emissions?

Re: The reason why we study and analyze other harmful gas emissions is that they all directly or indirectly impact GHG emissions. For example, the unburned ammonia will influence the thermal efficiency and therefore the GHG emissions per engine power, the NO_x emissions will limit the operating range and therefore affect the thermal efficiency and GHG emissions, and N₂O was also considered because its ultra-high global warming potential (i.e. a greenhouse effect around 298 times that of CO₂). We believe this kind question is very helpful for highlighting the disruptive nature of the IRGR ammonia combustion engine concept we proposed. As we reviewed in lines 213-234 of the revised manuscript, for the current state-of-the-art studies about "traditional" ammonia combustion engine, split diesel injection and very early injection have been used to reduce the unburned ammonia and therefore improve the thermal efficiency. However, all these studies face the bottleneck challenges such as trade-off relationship between unburned ammonia and NO_x emissions as well as high N₂O emissions. Since the advantages of hydrogen-enriched combustion and exhaust gas recirculation can be ingeniously combined, the IRGR technology proposed in this study can overcome the above bottleneck challenges faced by the state-of-the-art ammonia engine studies.

(Questions 3 and 6 will be answered together. Here, Question 4 first.)

4. The author completely neglected how to verify the emission reduction effect obtained in the paper.

Re: As we highlighted in Abstract of our previous draft, we use detailed 3D-CFD simulations that have undergone experimental validations under a wide range operating conditions to verify the potential of IRGR concept. We could not agree more that experimental validation is more convincing than 3D-CFD simulation validation. Unfortunately, very few institutions have the ability to conduct marine engine experiments owing to its large size and extremely high cost, especially for the ammonia-fueled marine engines that have just become a research hotspot in recent years. As a result, numerical simulation has been deemed as a necessary step at the early stage of any new engine development. Moreover, the IRGR ammonia engine concept was newly proposed in this work by us, and there is no similar ammonia engine available worldwide for experimental validation. In fact, we have already initiated the design and strength check of the IRGR engine. However, due to the relatively weaker manufacturing and system integration ability of universities compared to industries, even a moderate modification to the engine like the IRGR technology will usually take us more than 2 years and cost a large amount of money. Based on our recent successful experience in designing two "traditional" ammonia combustion engines (i.e. **Fig. R2** and **Fig. R6** in section "*Validation of 3D-CFD simulations against more experiment data*" in Pages 6-12 of this response file, the "traditional" here is compared to the new IRGR engine proposed in this study, it is also a dual fuel engine), we are confident in the successful operation of the IRGR engine in the future, but we believe that our current highly exciting 3D-CFD simulation results based on extensive experimentally verified high-precision numerical simulation model and robust chemical reaction mechanism will provide significant and invaluable inspirations to the industries who are also eagerly dedicated to the development of next-generation marine engines, thus collectively contributing to the achievement of the very urgent IMO GHG target.

We are very excited that the "Computational and simulation studies" are in the scope of Applied Science and Engineering Research @ Nature Communications, and we agree that only experimentally validated high-precision and convincing numerical studies will be considered. As you know, the ammonia-fueled engines have just become a research hotspot in recent years, and all the existing numerical simulations and chemical kinetic mechanism can only predict the engine performance and emissions data with an up to 60% ammonia energetic ratio to some extent, which is far behind the high-precision prediction of 90% ammonia energetic ratio and above required by the present study. As a result, before the study of IRGR concept, we dedicated our most efforts and time in the past two years to establish our 3D-CFD simulation models and detailed chemical kinetic mechanism consisting of 65 species and 344 reactions, and verify them against our state-of-the-art ammonia engine experimental data with ammonia energetic ratios of up to 90%. For the 3D-CFD simulation models,

the engine combustion chamber geometries were rigorously consistent with the experiment engine, and all the piston compression, pilot-diesel injection, spray break-up, evaporation, mixing of diesel spray and ammonia-air mixture, combustion, expansion and in-cylinder heat transfer processes have been well considered. As shown in the above section "*Validation of 3D-CFD simulations against more experiment data*" (i.e. Pages 6-12 in this response file), a total of 56 validation cases have been established and verified against the experimental data, and the predicted ignition delay, combustion phasing, heat release rate, in-cylinder pressure evolution and pollutant emissions including unburned ammonia, N₂O, NO_x and GHG agree very well with the experimental data under a wide range of ammonia energetic ratios (i.e. 40%, 60%, 80%, 90%) and engine loads (newly added in this revision, i.e. 50%, 75%, 85%, 100%), indicating the high-precision and robustness of our 3D-CFD numerical simulation models and detailed chemical reaction mechanism. Since all the important settings from the above experimentally verified 3D-CFD simulations such as engine specifications (i.e. engine bore diameter, stroke, combustion chamber geometries, and so on of the base engine), sub-models (i.e. spray, combustion, heat transfer, turbulence, heat transfer, and so on) and their relevant model constants, mesh methods and chemical reaction mechanism are rigorously continued to be used for the study of IRGR concept, we believe that the relevant calculated results are robust enough to support the highly exciting improvements by IRGR concept.

Moreover, to the best of our knowledge, we strongly believe that this is the first numerical simulations that have undergone experimental validation for pilot-diesel-ignited ammonia engine operating at ammonia energetic ratio exceeding 80%, and the accuracy and reliability of our CFD simulations are significantly higher than the state-of-the-art ammonia simulation studies. The newly developed detailed chemical reaction mechanism and thermo files have been added as Appendix 1 and 2 for further validation and use by academic and industrial sectors. Please review the section "*Validation of 3D-CFD simulations against more experiment data*" (i.e. Pages 6-12 in this response file) for more details, and we hope the above clarification and relevant experimental validations are helpful.

5. The limitation of this paper lies in why the author only chose ammonia as a new marine fuel for research. Is there any evidence that the next generation of marine engines will necessarily burn ammonia?

Re: Thanks a lot for raising this very interesting and invaluable issue. We would like to clarify that although we believe ammonia is the most promising alternative fuel for the future maritime sector according to our previous knowledge and the newly added comprehensive study "*Techno-economic study on various alternative marine fuels*" in Pages 22-44 of this response file also further confirms this point, we did not intend to convey that all next generation marine engines will necessarily use ammonia fuel. This is because that according to the current global annual production capacity, even

if all the Green and Blue alternative fuels including hydrogen, ammonia, methanol, methane were applicable in the maritime sector, they would be still far from enough to meet the large demands of global international ships, and more details can be found in the following section "*Techno-economic study on various alternative marine fuels*". In fact, we believe that before the fuel production capacity meets the demand, the maritime sector still needs an era of coexistence of multiple alternative fuels to mitigate the current severe shortage of Green and Blue alternative fuels. Considering currently around 99,248 international ships carry more than 80% of the global trade value and consume around 339 million tons of fuel oil annually, any alternative fuel, even if it holds a small market share, is of significant academic and economic value.

As for why we chose ammonia as a new marine fuel for research, here, we want to briefly discuss the advantages of using ammonia fuel in maritime sector. According to the following section "*Techno-economic study on various alternative marine fuels*", ammonia is the most competitive alternative fuel in a comprehensive consideration of economy, emissions, fuel availability, technology and safety. Especially, the annual fuel-related cost (i.e. the sum of fuel cost, on board fuel storage cost, engine and ship cost, extra voyage cost caused by the sacrifice of cargo capacity, GHG penalty by Well-to-Propeller CO₂ and CH₄ emissions, and GHG penalty by N₂O emissions from the current ammonia engine technology) of using ammonia fuel is significantly lower than that of using the other fuels (i.e. hydrogen, methanol and methane) in all fuel price scenarios and years (i.e. **Fig. R9** in Page 30). Compared to carbon-containing fuels like methanol and methane, the combustion of carbon-free ammonia fuel produces no engine-out emissions such as CO₂, PM, CO, HC, CH₄, formaldehyde etc., and the synthesis of ammonia does not require the expensive and immature direct air capture processes for obtaining CO₂, which corresponds to the low fuel production prices. Moreover, the use of ammonia fuel in maritime sector need not specially establish and monitor the complex carbon value chain compared to the use of carbon-containing fuels. Compared to the other carbon-free fuel hydrogen, the use of ammonia can significantly reduce the on board fuel storage cost and extra voyage cost as well as the total annual fuel-related cost. Moreover, the use of hydrogen as alternative fuel in maritime sector is currently limited by low technology readiness level (TRL) and safety problems, as shown in **Fig. R11** of section "*Techno-economic study on various alternative marine fuels*". Currently, in addition to the shortage of Blue and Green sources that all alternative fuels face, the most significant challenge in using ammonia as a marine fuel lies in engine combustion technology, specifically improving the combustion and thermal efficiency and further addressing N₂O emissions through disruptive engine technologies. As a result, the study on disruptive technology of ammonia marine engine is very valuable and urgent, and this is the reason why we primarily chose/focus on the ammonia combustion engine in recent years and this paper.

3. The paper does not have a clear research conclusion on which fuel to use in the future maritime sector, and the author did not provide useful conclusions through comparative research on different fuels, which is the biggest deficiency of this paper.

6. From the current development status of international maritime industry, it is still necessary to demonstrate which type of fuel, such as methanol, ammonia, or hydrogen, will be used as the fuel for the next generation of ship engines. If the author can comprehensively compare and study these fuels from various aspects such as emissions, economy, and safety, and provide clear conclusions, then this paper will be very valuable. However, at present, this paper is relatively subjective and does not conduct comprehensive research and comparison on the technical performance and feasibility of different fuels, which makes its academic value far from sufficient.

Response to the above two questions: Thanks a lot for your highly invaluable suggestions to improve the quality of this work. We totally agree that it is very important and invaluable to do a comprehensive comparison on various alternative fuels to demonstrate which fuel is more suitable for the next generation of marine engines. It should be noted that previous economic studies about alternative marine fuels mainly focus on the evolutions of fuel prices and ignore important issues such as on board storage cost, extra voyage cost caused by the sacrifice of cargo capacity, and GHG penalties. Inspired by your invaluable comments "*If the author can comprehensively compare and study these fuels from various aspects such as emissions, economy, and safety, and provide clear conclusions, then this paper will be very valuable*", during the revision processes, we have dedicated our most efforts to conducting a detailed techno-economic study on various alternative marine fuels to further improve the depth and value of this work, as shown in the following section "***Techno-economic study on various alternative marine fuels***" (i.e. Pages 22-44). In this techno-economic study, we conduct a comprehensive comparison of various alternative fuels (i.e. hydrogen, ammonia, methanol, methane) from five perspectives: economy, fuel availability, technology readiness level, safety and emissions. For the economic analysis, all the costs related to the use of alternative fuels have been considered as comprehensively as possible, including the fuel cost, fuel storage cost, engine and ship cost, extra voyage cost caused by the sacrifice of cargo capacity, GHG penalty by Well-to-Propeller CO₂ and CH₄ emissions, and GHG penalty by N₂O emissions from the current ammonia engine technology. More details can be found in the following section "***Techno-economic study on various alternative marine fuels***". To avoid too long for the main paper, the full "***Techno-economic study on various alternative marine fuels***" is attached as Appendix 4 to the revised manuscript, and the main conclusions of this Techno-economic study are summarized in the first section of the Introduction (i.e. Section 1.1) of the revised manuscript to further improve the depth and value of this paper. We hope this Techno-economic study are acceptable and helpful, and thank you for your kind suggestion again. Please kindly review section "***Techno-economic study on various alternative marine fuels***" in Pages 22-44 of this response file for more details, and the main conclusions are

summarized as follows:

- In terms of economy, in 2020, the annual fuel-related cost of using Green fuels is significantly higher than that of Blue fuels. However, in 2050, the price advantage of Blue fuels will be offset by the reduced Green fuel price and GHG emission penalties to some extent, resulting in the annual fuel-related cost of using Blue fuels being close to that of using Green fuels. As a result, owing to the significant price advantage, the Blue fuels that feature around 80% reduction in WtP CO₂ and CH₄ emissions (i.e. near-zero emissions) will serve as an important transitional fuel between fossil fuels and Green fuels in the next 30 years. Owing to the highest fuel storage cost and extra voyage cost, the use of hydrogen will result in the highest annual fuel-related cost. Due to the most expensive fuel cost, the annual fuel-related cost of using methanol fuel ranks second, only after the hydrogen fuel. Since the use of ammonia fuel features the low fuel cost and fuel storage cost as well as low extra voyage cost, its annual fuel-related cost is significantly lower than the other fuels in all fuel price scenarios and years, even the GHG penalty by N₂O emissions having been totally considered.
- The economic advantages of using ammonia fuel in the future maritime sector will be more attractive if the N₂O emissions can be addressed. Taking the low fuel price scenario as an example, if the N₂O emissions from ammonia combustion engines can be solved by disruptive engine technologies, the annual fuel-related cost of Blue ammonia will be 57%, 60%, 62%, 66% of the cost of Blue methanol in 2020, 2030, 2040 and 2050, respectively. For the Green fuel in 2050, the annual fuel-related cost of ammonia is 82% of the cost of methanol with the consideration of economic penalty from N₂O emissions, while this value is reduced to 64% if the N₂O emissions from current ammonia engines can be solved. Moreover, if the N₂O emissions from the current ammonia combustion engine can be addressed, in 2050, the global international ships will save around 149 billion USD per year in GHG penalty from N₂O emissions. The above discussion about N₂O emissions highlights the economic advantages of using ammonia fuel in the future maritime sector and the necessity of the relevant disruptive engine technologies.
- In terms of technology readiness, the use of methane fuel has the same bunkering and engine TRL with fuel oil. The technological maturity of marine methanol engines can be achieved (i.e. 2022~2024) approximately 2 years ahead of that of marine ammonia engines (i.e. 2025~2026), but they can be considered to achieve technological maturity (i.e. TRL 9) at the same time in terms of the net-zero target in 2050. Correspondingly, the fuel quality standards for methanol fuel (i.e. TRL 3) are also slightly ahead of that for ammonia fuel (i.e. TRL 2), while the TRL of bunkering equipment is the same for ammonia and methanol fuel (i.e. TRL 7). The use of hydrogen fuel has the lowest bunkering and engine TRL, and this is mainly due to the economic and safety concerns.

- Flash point, minimum flammability limit in air, toxicity and corrosiveness are considered in the safety evaluation of different fuels. Methane has similar safety characteristics to fuel oil except for its extremely low flash point, while the low flash point and low flammability limit of hydrogen pose challenges in terms of safety. The poor combustion characteristics of ammonia fuel is the largest challenge for engine technology, but it is actually an advantage from the perspective of fire hazard and safety. Although ammonia and methanol are toxic and corrosive, they have been safely transported by international ships as chemical cargoes for a long time, demonstrating that the safety management of these two fuels is feasible and mature. As a result, the necessary safety regulations of on board use are expected to be ready for ammonia fuel within 5 years.
- In terms of emissions, the Well-to-Propeller GHG emissions (CO₂ and CH₄) and engine-out N₂O emissions have been penalized and discussed in detail in the economic study. Due to the absence of carbon (C) and sulfur (S) elements, the use of hydrogen (H₂) and ammonia (NH₃) fuels will not produce engine-out emissions such as CO₂, SO_x, particulate matter (PM), CO, hydrocarbon (HC) and formaldehyde (HCHO), making these two alternative fuels most attractive from a long-term perspective. Methanol (CH₃OH) and methane (CH₄), as carbon-containing fuels, exhibit similar engine-out emission characteristics (i.e. CO₂, PM, CO, HC, CH₄, formaldehyde etc.). Compared to the use of fuel oil, the use of methanol and methane can reduce PM emissions to some extent, but it will lead to a significant increase in unregulated harmful emissions like formaldehyde.
- As for the fuel availability, the global annual production of hydrogen is the highest and around 80% higher than that of ammonia, while the annual production of ammonia is approximately twice that of methanol. If all the considered alternative fuels were to be used in the maritime sector, they could meet approximately 78% of the total energy requirement. However, most of these alternative fuels produced to date are Gray and Brown fuels, highlighting the production capacity of Green and Blue fuels needs to be significantly increased in the near future to meet the IMO target of net-zero emissions in 2050.
- In summary, ammonia is the most competitive alternative fuel in a comprehensive consideration of economy, emissions, fuel availability, technology and safety. Especially, the annual fuel-related cost of using ammonia fuel is significantly lower than that of using the other fuels in all fuel price scenarios and years, even the GHG penalty by N₂O emissions having been totally considered. Compared to carbon-containing fuels like methanol and methane, the combustion of carbon-free ammonia fuel produces no engine-out emissions such as CO₂, PM, CO, HC, CH₄, formaldehyde etc., and the synthesis of ammonia does not require the expensive and immature direct air capture processes for obtaining CO₂, which corresponds to the low fuel production prices. Moreover, the

use of ammonia fuel in maritime sector need not specially establish and monitor the complex carbon value chain compared to the use of methanol and methane fuels. Currently, in addition to the shortage of Blue and Green sources that all alternative fuels face, the most significant challenge in using ammonia as a marine fuel lies in engine combustion technology, specifically achieving technological maturity as soon as possible (i.e. around 2025~2026) and further addressing N₂O emissions through combustion optimizations and disruptive technologies. As carbon-containing alternative fuels, methanol and methane exhibit similar economy and emission characteristics and can be compared together. For example, using methanol and methane can eliminate SO_x emissions and reduce PM emissions, but it is still inevitable to produce NO_x, CO and HC emissions as well as a large amount of unregulated harmful emissions like formaldehyde. The bunkering and engine TRL of methane are higher than those of methanol, and the transport of methane can share the mature infrastructure of natural gas, which can help expand the annual production capacity of methane. However, the methanol is superior than methane in terms of on board storage cost and methane slip. The present techno-economic study also specifically compare ammonia fuel and methanol fuel, and more details are summarized in **Fig. R11**. Hydrogen is the cleanest fuel, but its application in maritime sector is currently limited by issues such as high fuel-related cost, low TRL and safety problems.

- According to the current global annual production capacity, even if all the Green and Blue alternative fuels were applicable in the maritime sector, they would still far from enough to meet the large demands of global international ships. As a result, although ammonia has been considered as the most promising alternative marine fuel according to the results of the present study, the maritime sector still needs an era of coexistence of multiple alternative fuels to mitigate the current severe shortage of Green and Blue fuels. Considering the fuel-related cost of global international ships in 2050 calculated to be between 635 billion USD (i.e. the use of Blue ammonia in the low fuel price scenario) and 1,597 billion USD (i.e. the use of Green hydrogen in the high fuel price scenario) per year, any alternative fuel, even if it holds a small market share, is of significant academic and economic value. The above highlights the necessity and urgency of developing disruptive engine technologies to improve fuel efficiency and reduce emissions.

4. To Reviewer #2: Techno-economic study on various alternative marine fuels

(Attached as Appendix 4 of the revised manuscript)

1. Economic analysis on various alternative marine fuels

1.1. Alternative marine fuels considered in this study

According to the latest definition in 2023 by Lloyd's Register (LR) [1], "zero greenhouse gas (GHG) emissions in maritime sector" can be divided into: (1) Absolute-zero emissions mean that there are no GHG emissions from the whole Well-to-Propeller (WtP) lifecycle; (2) Net-zero emissions can be achieved when anthropogenic GHG emissions to atmosphere can be balanced by anthropogenic removal from the perspective of WtP lifecycle; (3) Near-zero emissions refer to a reduction of more than 80% in GHG emissions compared to low-sulphur fuel oil (LSFO). On July 7, 2023, the International Maritime Organization (IMO) proposed a reduction of at least 70% in GHG from international ships by 2040, with the ultimate goal of achieving net-zero GHG shipping by around 2050 [2]. As a result, replacing the marine oil fueled diesel engines, which account for around 98.8% of the prime movers in international ships [3], with alternative fuel fueled engines that feature the near-zero or net-zero GHG emissions is inevitable to achieve the IMO target in the next 30 years.

Ammonia [4-6], hydrogen [7-9], methanol [10-12] and methane [13] have been considered as potential alternative fuels for the future maritime sector. According to the fuel production routes, these alternative fuels are generally divided into: (1) Gray/Brown fuels, utilizing natural gas/coal as feedstocks, and no carbon capture technology is used in the fuel production pathway; (2) Blue fuels, still using fossil fuel as feedstocks, but the CO₂ emissions during the fuel production processes are captured and permanently stored; (3) Green fuels that are produced from renewable energy sources such as wind or solar power [3, 8, 14, 15]. The IMO highlights that decarbonizing the global shipping should not shift emissions to other sectors [3]. As a result, although most of these alternative fuels produced to date are Gray and Brown fuels, the maritime sector primarily focuses on the use of Blue/Green fuels that feature near-zero/net-zero WtP CO₂ and CH₄ emissions [3, 8, 15]. It should be noted that if the focus is upgraded to absolute-zero emissions and the onboard carbon capture is not used, any fuel contains carbon such as methanol and methane should be precluded, which means that only the fuels that contain no carbon like hydrogen and ammonia can be considered as candidates.

The present study summarizes the fuel production and supply pathways of 7 alternative marine fuels (i.e. Green hydrogen, Blue hydrogen, Green ammonia, Blue ammonia, Green methanol, Blue methanol and Green methane) widely discussed in the current maritime sector into a conceptual diagram, as shown in **Fig. R8**. These 7 alternative fuels are the focus of this economic study, and the fuel oil is also added for comparison. The Blue methane is not considered in the industry and maritime sector [13] as well as in the present study, and this is because this is an unfeasible fuel production route as the feedstocks (i.e. natural gas) and synthetic product (i.e. methane) are almost similar. Main considerations of the fuel production pathways in **Fig. R8** are summarized as follows:

Fig. R8. Conceptual illustration of the fuel production and supply pathways of 7 alternative marine fuels widely discussed in the current maritime sector and considered in this study.

- Feedstocks: For the Blue fuel, the natural gas is used as feedstocks, and this is consistent with the latest available fuel price data from Maersk [4], American Bureau of Shipping (ABS) [8], Det Norske Veritas (DNV) [3] and Lloyd’s Register (LR) [16]. For the Green fuel, the electricity from renewable wind or solar is employed as feedstocks. The biomass is not considered as feedstocks and fuels in the present study owing to the challenge of large-scale production for maritime sector [17, 18]. Another reason for not considering the biomass is its wide variety such as fatty-acid methyl ester (FAME) biodiesel, Fischer-Tropsch (FT) diesel, hydrotreated renewable diesel, straight vegetable oil (SVO) and dimethyl ether (DME), leading to significant differences in fuel prices and Well-to-Propeller emissions [18, 19].
- Hydrogen production: For the Blue hydrogen, it is produced from natural gas using the steam methane reforming (SMR), and the carbon emissions during the SMR processes should be captured and permanently stored [3, 20]. For the Green hydrogen, the renewable electricity is used to power the water electrolysis processes with no carbon emissions.
- Ammonia, methanol and methane production: The Blue/Green ammonia is produced through the well-established Haber-Bosch synthesis processes, and the Blue/Green hydrogen and nitrogen from air separation are used as input for ammonia synthesis. For the Blue/Green methanol and methane synthesis processes, the Blue/Green hydrogen and CO₂ from direct air capture (DAC) are used as input. It should be noted that according to LR’s definition in 2023 [14] and ABS’s

definition in 2022 [8], two methods include direct air capture and bioenergy with carbon capture (BECCS) can be recognized as reasonable sources of CO₂ for the Green/Blue methanol synthesis processes. However, the cost of BECCS is currently unclear and the large-scale production of biomass presents challenges [17, 18], so the maritime sector and present study primarily focus on the DAC when evaluates the fuel price, and more details can be found in Section 1.4.

- Marine fuel supply: In the present economic study, all the fuels are assumed to be stored and supplied to ships as the liquid phase, which can ensure the minimum sacrifice of cargo capacity.
- Ships: Since the Blue/Green fuels that feature near-zero/net-zero Well-to-Propeller carbon emissions are used in the present economic study, the expensive and low technology maturity onboard carbon capture technology [21-23] is not further considered in this study.

Table R1 summarizes and calculates the Well-to-Propeller GHG emissions (CO₂ and CH₄) of 7 alternative fuels in **Fig. R8** based on the latest data from LR [16], Maersk [4, 13, 15] and ABS [8]. It can be seen that only the Green hydrogen, Green ammonia, Green methanol can be considered as net-zero alternative fuels from the perspective of Well-to-Propeller GHG emissions (CO₂ and CH₄). For the Green methane, the Well-to-Propeller GHG emissions are around 5.8% of fuel oil, and this can be attributed to the methane slip during the fuel production and engine exhaust processes. For the Blue hydrogen, Blue ammonia and Blue methanol, the Well-to-Propeller GHG emissions (CO₂ and CH₄) are around 17.5% to 19.6% of fuel oil, and this can be attributed to the limit of carbon capture ability and methane slip during the fuel production process. The averaged data in **Table R1** will be used in the following techno-economic studies.

Table R1. Well-to-Propeller GHG emissions (CO₂ and CH₄) of 7 alternative fuels widely discussed in the current maritime sector and considered in this study

Fuel type	LR [16] (gCO ₂ -eq/MJ)	Maersk (gCO ₂ -eq/MJ)	ABS [8] (gCO ₂ -eq/MJ)	Average (gCO ₂ -eq/MJ)	Definition [1]
Low-sulphur fuel oil	81.4	96.0 [15]	96.0	91.1 (100%)	-
Green hydrogen	0.0	1.0 [15]	1.0	0.7 (0.8%)	^d Net- & Absolute-zero
^a Green ammonia	0.0	1.0 [4]	1.0	0.7 (0.8%)	^d Net- & Absolute-zero
Green methanol	0.0	1.0 [15]	1.0	0.7 (0.8%)	^d Net-zero
Green methane	0.0	^b 10.0 [13]	5.8	5.3 (5.8%)	Near-zero
Blue hydrogen	15.0	16.3 [15]	16.3	15.9 (17.5%)	Near-zero
^a Blue ammonia	15.0	19.0 [4]	18.2	17.4 (19.1%)	Near-zero
^c Blue methanol	17.0	18.3	18.3	17.9 (19.6%)	Near-zero

^a For ammonia combustion engines, the GHG emissions of N₂O are not considered in Refs. [4, 8, 16] and **Table R1**, but this will be considered in the present techno-economic study, and more details can be found in Section 1.2.

^b According to Maersk, the Well-to-Propeller GHG emissions (CO₂ and CH₄) of Green methane is 4-

16 gCO₂-eq/MJ, and the averaged value is used here.

^c The data of Blue methanol is not given in the references. According to **Fig. R8**, the difference in Well-to-Propeller GHG emissions (CO₂ and CH₄) between Blue methanol and Green methanol only arises from the production of Blue hydrogen and Green hydrogen, and therefore the Well-to-Propeller GHG emissions (CO₂ and CH₄) of Blue methanol can be derived from the existing data.

^d 0.7 gCO₂-eq/MJ is ignored here.

1.2. Methods

In the present economic analysis of different alternative marine fuels, a 46,665 dead weight tonnage (DWT) bulk carrier is used as the background ship, and the ship technical and operational specifications are given in **Table R2**. All the cost related to the use of alternative fuels have been considered as comprehensively as possible, including the fuel cost, fuel storage cost, engine and ship cost, extra voyage cost owing to the low energy density of the alternative fuels and the reduced cargo capacity, GHG penalty by Well-to-Propeller CO₂ and CH₄ emissions, and GHG penalty by N₂O emissions from the current ammonia engine technology, more details are summarized as follows:

Table R2. Ship technical and operational specifications

Ship type	Bulk carrier
Deadweight (tonnage)	46,665
Displacement (tonnage)	59,481
Cruising distance (nautical mile)	10,000
Engine rated power (kW)	7,948
Engine usual power (kW)	7,153
Total cargo capacity (m ³)	52,848
Fuel bunker capacity (m ³)	1,180
Days at sea per year	250

- Fuel cost: The present study summarizes the price of the alternative fuels and fuel oil in **Fig. R8** from the latest data of Maersk [4, 10, 13, 24], ABS [8], DNV [3] and LR [16], and derives the low fuel price scenario, medium fuel price scenario and high fuel price scenario, respectively. More details can be found in **Fig. R10** of Section 1.4. Moreover, when calculating the annual fuel cost, it is assumed that the ship operates at sea for 250 days per year.
- Fuel storage cost: This means the capex cost of on board fuel storage tank. According to LR and International Energy Agency (IEA), the storage capex is 0.14 USD/kg for diesel fuel [25], 0.144 USD/kg for methanol fuel [25], 0.7 USD/kg for ammonia fuel [16], 7.14 USD/kg for liquid methane fuel [16], and 333 USD/kg for liquid hydrogen fuel [26, 27]. The lifespan of fuel storage tank is assumed to be 10 years for ammonia fuel, liquid methane fuel and liquid hydrogen fuel, which is consistent with the assumption by LR [16]. For the diesel and methanol fuels, the lifespan

of the fuel storage tank is assumed to be consistent with that of engine and ship (i.e. 30 years). Moreover, for the storage of liquid hydrogen and liquid methane, the fuel loss by boil-off stream and cost for keeping the hydrogen and methane in cryogenic condition is not considered.

- Engine and ship cost: In the calculation processes, the engine capex cost is assumed to be 400 USD/kW for diesel fuel, while it is assumed to be 590 USD/kW for the alternative fuels [16]. Excluding the capex cost of fuel storage and engine, the cost of ship is assumed to be 25 million USD for all the fuels. Moreover, the lifespan of the ship and engine is assumed to be 30 years.
- GHG penalty by Well-to-Propeller CO₂ and CH₄ emissions: This part was totally neglected in the previous economic analyses. Indeed, in the current maritime sector, the carbon tax has been considered and widely discussed as an effective method to achieve the target of net-zero in 2050 [3, 8]. For example, ABS reported that the average carbon tax should be around 191 USD per ton of equivalent CO₂ (USD/tCO₂-eq) to decarbonize the international ships by 2050 [8]. DNV reported that the carbon tax should be as high as 250 USD/tCO₂-eq to achieve the aim of decarbonization by 2050 [3]. As shown in **Table R1**, only the Green hydrogen, Green ammonia, Green methanol can be considered as net-zero (i.e. WtP CO₂ and CH₄) alternative fuels, and the other fuels will incur the carbon tax penalties. Refer to the estimated carbon tax by DNV [3], this study assumes a gradually increasing carbon tax, specifically 83 USD/tCO₂-eq in 2030, 167 USD/tCO₂-eq in 2040 and 250 USD/tCO₂-eq in 2050.
- GHG penalty by N₂O emissions: This part was also entirely ignored in the previous economic analyses. According to a recent Comment in Nature Energy [28], the N₂O emissions of the current ammonia combustion engine is around 1 g/kWh. Since N₂O has a greenhouse effect around 298 times that of CO₂, the economic penalty of N₂O emissions is also considered in the present economic analysis.
- Extra voyage cost: The energy density is 35.7 MJ/L for diesel, 23.0 MJ/L for liquid methane, 15.9 MJ/L for methanol, 11.3 MJ/L for liquid ammonia and 8.5 MJ/L for liquid hydrogen [29], which leads to the need of larger fuel volume for the alternative fuels. Moreover, if the additional equipment such as that for maintaining the cryogenic state of liquid hydrogen and liquid methane is further considered, the fuel and tank volumes for methanol, liquid methane, liquid ammonia and liquid hydrogen will be 2.2 times, 2.5 times, 3.1 times and 6.7 times that of diesel fuel, respectively [29]. For the background ship, the fuel and tank volume is 1,475 m³ for the diesel fuel, and the total cargo capacity is 52,848 m³. If the liquid hydrogen is used as the alternative fuel, there will be a loss of around 16% of the total cargo capacity (i.e. 8,413 m³), leading to around 19% extra voyage cost to achieve the equal annual cargo transport capacity compared to that using fuel oil. Unfortunately, the extra voyage is usually ignored in the previous economic study, and this will be fully considered in this study. It should be noted that when assessing the impact of alternative fuels on cargo capacity of bulk carrier, both the volume and weight of the

fuel and fuel tank should be considered. In the calculation of the extra voyage cost, the present economic study primarily considers the loss of cargo volume capacity resulting from the use of alternative fuels, which is more relevant for other vessel types such as container ships.

1.3. Main results and discussions

Fig. R9 shows the annual fuel-related cost when using different alternative fuels, and the technical and operational specifications of the background ship are given in **Table R2**. While most of the previous economic studies about alternative marine fuels mainly focus on the evolutions of fuel prices [30], the present study considers all the cost related to the use of alternative fuels as comprehensively as possible, including the fuel cost, fuel storage cost, engine and ship cost, extra voyage cost caused by the sacrifice of cargo capacity, GHG penalty by Well-to-Propeller CO₂ and CH₄ emissions, and GHG penalty by N₂O emissions from the current ammonia engine technology. Since the IMO proposed a reduction of at least 70% in GHG from international ships by 2040, the annual fuel-related cost of fuel oil is no longer provided for 2040 and 2050.

In terms of fuel cost, all alternative fuels are significantly more expensive than fuel oil. Until 2050, the costs of Green fuels are obviously higher than those of Blue fuels, and this is mainly because that the renewable electricity as feedstock is far more expensive than natural gas. Since the price of renewable electricity is expected to significantly decrease in the next 30 years owing to the large-scale production and increased technological maturity, the cost difference between Blue and Green fuels will gradually decrease from 2020 to 2050. For example, in 2020, the cost of Green hydrogen is around twice that of Blue hydrogen, and this value reduces to 1.5 times in 2050. When comparing fuels within the same color, the use of hydrogen leads to the lowest fuel cost even with the consideration of the expensive cryogenic liquefaction processes. As shown in **Fig. R8**, this is because the production pathway for hydrogen fuel is simpler compared to the synthesis of ammonia, methanol and methane fuels. The fuel costs of methanol and methane are significantly higher than that of ammonia, and this is mainly because the cost of directly capturing CO₂ from the air is much higher than separating nitrogen from the air. Here, it needs to be noted again that according to LR's definition in 2023 [14] and ABS's definition in 2022 [8], only DAC and BECCS can be recognized as reasonable sources of CO₂ for the Green/Blue methanol and methane synthesis processes. Since the cost of BECCS is currently unclear and the large-scale production of biomass presents challenges [17, 18], the maritime sector and present study primarily focus on the DAC when evaluates the fuel cost.

The engine and ship cost is relatively small, especially for the alternative fuels. For example, for the high fuel price scenario in 2020, the engine and ship cost accounts for approximately 27% of the total annual cost for fuel oil, while this cost is less than 3% and 6% for the use of Green fuels and Blue fuels, respectively. As discussed in Section 1.2, the on board fuel storage tank capex cost of hydrogen is significantly higher than those of the other alternative fuels. As a result, compared to the most expensive methanol fuel, the significant price advantage of hydrogen fuel is almost completely

offset by the high fuel storage cost. Moreover, when further considering the highest extra voyage cost caused by the largest sacrifice of cargo capacity, the hydrogen fuel has the highest annual fuel-related cost among all the alternative fuels. It should be noted that for the storage of liquid hydrogen and liquid methane, the fuel loss by boil-off stream and cost for keeping the hydrogen and methane in cryogenic condition is not considered, and the annual fuel-related cost will further increase if the above issues are fully considered.

In the current maritime sector, the carbon/GHG tax has been considered and widely discussed as an effective method to achieve the net-zero target in 2050 [3, 8], and it is also considered in this study. As shown in **Table R1**, only the Green hydrogen, Green ammonia, Green methanol can be considered as net-zero Well-to-Propeller GHG emissions (CO_2 and CH_4), and the other fuels will incur the carbon tax penalties. For the use of ammonia fuel, since engine-out N_2O emissions have a greenhouse effect around 298 times that of CO_2 [28], the economic penalty of N_2O emissions from the current ammonia combustion engine is also considered. As shown in the low fuel price scenario, in 2020, the annual fuel-related cost of using Green fuels is significantly higher than that of Blue fuels. However, in 2050, the price advantage of Blue fuels will be offset by GHG emission penalties to some extent, resulting in the annual fuel-related cost of using Blue fuels being close to that of using Green fuels.

In summary, owing to the significant price advantage, the Blue fuels that feature around 80% reduction in WtP CO_2 and CH_4 emissions (i.e. near-zero emissions) will serve as an important transitional fuel between fossil fuels and Green fuels in the next 30 years. Owing to the highest fuel storage cost and extra voyage cost, the use of hydrogen will result in the highest annual fuel-related cost. Due to the most expensive fuel cost, the annual fuel-related cost of using methanol fuel ranks second, only after the hydrogen fuel. Since the use of ammonia fuel features the low fuel cost and fuel storage cost as well as low extra voyage cost, its annual fuel-related cost is significantly lower than the other fuels in all fuel price scenarios and years, even the GHG penalty by N_2O emissions have been totally considered. It should be noted that for different fuel price scenarios in 2050, the GHG penalty by N_2O emissions can be as high as 24% of the total annual cost when using ammonia fuel. As a result, the in-cylinder reforming gas recirculation technology that features more than 90% N_2O emissions reduction proposed in this study will further help highlight the economic advantages of using ammonia fuel in the future maritime sector. Taking the low fuel price scenario as an example, if the N_2O emissions from ammonia combustion engines can be solved, the annual fuel-related cost of Blue ammonia will be 57%, 60%, 62%, 66% of the cost of Blue methanol in 2020, 2030, 2040 and 2050, respectively. For the Green fuel in 2050, the annual fuel-related cost of ammonia is 82% of the cost of methanol with the consideration of N_2O emissions, while this value is reduced to 64% if the N_2O emissions from current ammonia engines can be solved. More specifically, if the N_2O emissions from the current ammonia combustion engine can be addressed, in 2050, the global international ships will save around 149 billion USD per year in GHG penalty from N_2O emissions.

Low Fuel Price Scenario

(a) Low fuel price scenario

Medium Fuel Price Scenario

(b) Medium fuel price scenario

(c) High fuel price scenario

Fig. R9. Annual fuel-related cost when using different alternative fuels. (a) Low fuel price scenario; (b) Medium fuel price scenario; (c) High fuel price scenario. The GHG penalty by N_2O emissions from current ammonia engine have been totally considered here, and the economic advantages of using ammonia fuel in the future maritime sector will be more attractive if the N_2O emissions can be addressed.

1.4. Supporting data of economic analysis

Fig. R10 shows the prices of 7 alternative fuels and fuel oil from the latest data of Maersk [4, 10, 13, 24], ABS [8], DNV [3] and LR [16]. Three issues need to be clarified during the data collection and calculation process: Firstly, it should be noted that as discussed in the above section, both LR [14] and ABS [8] highlight that the direct air capture (DAC) or bioenergy with carbon capture (BECCS) should be employed to obtain CO_2 for Green methanol and Green Methane synthesis processes. However, ABS misused the fossil CO_2 pathway rather than DAC or BECCS CO_2 pathway, and the price difference in CO_2 sources between fossil pathway and DAC pathway [10, 13] has been considered in the present study when summarizing ABS's Green methanol and Green methane data. As shown in **Fig. R10e** and **Fig. R10g**, the corrected ABS data aligns better with the data of LR, Maersk and DNV. Secondly, the fuel price of Blue methanol is not given in the references. According to the fuel production pathways summarized in **Fig. R8**, the difference in fuel price between Blue methanol and Green methanol only arises from the production of Blue hydrogen and Green hydrogen, and therefore the fuel price of Blue methanol can be derived from the existing data. Thirdly, DNV

gave a low fuel price scenario and a high fuel price scenario in 2050, and the averaged value is summarized in Fig. R10.

Fig. R10. Fuel price from the latest data of Maersk [4, 10, 13, 24], ABS [8], DNV [3] and LR [16].

As shown in Fig. R10, until 2050, Green fuels that feature net-zero WtP CO₂ and CH₄ emissions are more expensive than Blue fuels characterized by near-zero WtP emissions, and this is mainly because that the renewable electricity as feedstock is far more costly than natural gas. Since the price of renewable electricity is expected to significantly decrease in the next 30 years owing to the large-scale production and increased technological maturity, the price difference between Blue and Green fuels will gradually decrease in the next 30 years. For example, according to LR's data, in 2020, the price of Green hydrogen is around twice that of Blue hydrogen, and this value reduces to 1.5 times in 2050. As a result, from the perspective of fuel price, Blue fuels that feature around 80% reduction in WtP CO₂ and CH₄ emissions will serve as an important transitional fuel between fossil fuels and Green fuels in the next 30 years. The data from LR, ABS, DNV, Maersk all indicates that the fuel prices of methanol and methane are significantly higher than ammonia, this is mainly because the cost of directly capturing CO₂ from the air is much higher than separating nitrogen from the air.

Moreover, the price of fuel oil is also summarized in **Fig. R10h**, it can be seen that the prices of all alternative fuels are significantly higher than that of fuel oil. Generally, the LR provides the most comprehensive fuel price data, and it aligns with the limited data provided by DNV, ABS and Maersk. As a result, this economic analysis uses LR's fuel price data, and averages the low fuel price scenario and high fuel price scenario to obtain the medium fuel price scenario.

2. Technical, safety and emission analysis on various alternative marine fuels

2.1. Main results and discussions

Fig. R11 summarizes the comparison of fuel availability, technology readiness level (TRL), safety and emissions among different fuels for maritime sector, and the supporting data is given in Section 2.2 to Section 2.5. In terms of technology readiness, it can be seen that the use of methane fuel has the same bunkering and engine TRL with fuel oil. This is because natural gas, which has similar physicochemical properties to methane, has already been used as a marine fuel for more than 20 years. The technological maturity of marine methanol engines can be achieved (i.e. 2022~2024) approximately 2 years ahead of that of marine ammonia engines (i.e. 2025~2026), but they can be considered to achieve technological maturity (i.e. TRL 9) at the same time in terms of the net-zero target in 2050. Correspondingly, the fuel quality standards for methanol fuel (i.e. TRL 3) are also slightly ahead of that for ammonia fuel (i.e. TRL 2), while the TRL of bunkering equipment is the same for ammonia and methanol fuel (i.e. TRL 7). The use of hydrogen fuel has the lowest bunkering and engine TRL, and this is mainly due to the economic and safety concerns.

Flash point, minimum flammability limit in air, toxicity and corrosiveness are all critical factors to consider in the safety evaluation of different fuels, and the above information is summarized in **Table R6** to **Table R9** of Section 2.4. Methane has similar safety characteristics to fuel oil except for its extremely low flash point, while the low flash point and low flammability limit of hydrogen pose challenges in terms of safety. Although the poor combustion characteristics of ammonia fuel is the largest challenge for engine technology, it is actually an advantage from the perspective of safety. It can be seen that the ammonia has the highest flash point and minimum flammability limit in air, making it the safest alternative fuel from the perspective of fire hazard. Although ammonia and methanol are toxic and corrosive, they have been safely transported by international ships as chemical cargoes for a long time, demonstrating that the safety management of these two fuels is feasible and mature. As a result, DNV predicts that all the necessary safety regulations of on board use will be ready for ammonia fuel within 5 years, as shown in **Table R10** of Section 2.4.

In terms of emissions, the WtP GHG (CO₂ and CH₄) and engine-out N₂O emissions have been penalized and discussed in detail in the above economic study, and so no details here to avoid repeat. Due to the absence of carbon (C) and sulfur (S) elements, the use of hydrogen (H₂) and ammonia (NH₃) fuels will not produce engine-out emissions such as CO₂, SO_x, particulate matter (PM), CO,

hydrocarbon (HC) and formaldehyde (HCHO), and this is one of the reasons why these two alternative fuels have attracted the widespread attention in the maritime sector. Methanol (CH₃OH) and methane (CH₄), as carbon-containing fuels, exhibit similar engine-out emission characteristics (i.e. CO₂, PM, CO, HC, CH₄, formaldehyde etc.). Compared to the use of fuel oil, the use of methanol and methane can reduce PM emissions to some extent, but it will lead to a significant increase in unregulated harmful emissions like formaldehyde.

Fig. R11. Comparison of fuel availability, technology, safety and emission among different fuels for maritime sector. (*The fuel availability of methane is defined as dependent, details in Section 2.2)

As for the fuel availability, the global annual production of hydrogen is the highest and around 80% higher than that of ammonia, while the annual production of ammonia is approximately twice that of methanol. As shown in **Table R3**, based on the current global production capacity, if all the considered alternative fuels (i.e. hydrogen, ammonia, methanol, methane) were to be used in the maritime sector, they could meet approximately 78% of the total energy requirement. However, most of these alternative fuels produced to date are Gray and Brown fuels. For example, the annual

production of Green methanol is only around 0.2% of the total methanol production [31]. As a result, since the IMO highlights that decarbonizing the global shipping should not shift emissions to other sectors, the production capacity of Green and Blue fuels needs to be significantly increased in the near future to meet the IMO target of net-zero emissions in 2050.

In summary, ammonia is the most competitive alternative fuel in a comprehensive consideration of emissions, fuel availability, technology and safety. Currently, in addition to the shortage of Blue and Green sources that all alternative fuels face, the most significant challenge in using ammonia as a marine fuel lies in engine combustion technology, specifically achieving technological maturity as soon as possible (i.e. around 2025~2026) and further addressing N₂O emissions through combustion optimizations and disruptive technologies. As carbon-containing alternative fuels, methanol and methane exhibit similar emission characteristics and can be compared together. For example, using methanol and methane can eliminate SO_x emissions and reduce PM emissions, but it is still inevitable to produce NO_x, CO and HC emissions as well as a large amount of unregulated harmful emissions like formaldehyde. Since natural gas that features similar physicochemical properties to methane has already been used as a marine fuel for more than 20 years, the bunkering and engine TRL of methane are higher than those of methanol. Moreover, the transport of methane can share the mature infrastructure of natural gas, which can help expand the annual production capacity of methane. However, the methanol is superior than methane in terms of on board storage and methane slip. The present techno-economic study also specifically compare ammonia fuel and methanol fuel, and more details are summarized in **Fig. R11**. Hydrogen is the cleanest fuel, but its application in maritime sector is currently limited by issues such as TRL, safety and onboard storage.

2.2. Supporting data of fuel availability

Table R3 summarizes the comparison of the fuel availability of different fuels for maritime sector. The assessment here is primarily based on the current global annual production of these alternative fuels, which can reflect the current capacity of relevant production and transportation infrastructures and provide an objective evaluation. As shown in **Table R3**, around 100,000 international ships in current maritime sector need 339 million tonnes fuel oil per year, and the total energy of these fuel oils is used as the baseline to evaluate the fuel availability of the considered alternative fuels. It should be noted that the natural gas is the most important feedstock for the current hydrogen, ammonia and methanol production, but as discussed in the above section, it is not used as feedstock for methane production because this is an unfeasible production pathway (i.e. the feedstock is similar to the synthetic product). As a result, the renewable mass is the main feedstock for methane synthetic to date, and the annual production of renewable methane can only be classified as "Level 1". However, the fuel availability of methane is classified as "Dependent" in the present study, this is because that the transport of methane can share the mature infrastructure of natural gas, which is helpful for expanding the annual production capacity of methane.

Table R3. Comparison of the fuel availability of different fuels for maritime sector (The annual total energy of fuel oil in current maritime sector is used as the baseline)

Fuel type	Current annual production (million tonnes)	Annual total energy (GJ)	Relative value (%)	Level
Fuel oil	339 (For maritime) [32]	14,407,500,000	100	10
Hydrogen	50.0 [33]	6,000,000,000	41.6	5
Ammonia	176.0 [33]	3,308,800,000	23.0	3
Methanol	85.0 [33]	1,691,500,000	11.7	2
Methane	5.3 [34]	202,827,840	1.8	^a Dependent

^a Currently, the renewable mass is the main feedstock for methane synthetic, and the annual production of renewable methane can only be classified as "Level 1". However, the fuel availability of methane is classified as "Dependent" here, this is because that the transport of methane can share the mature infrastructure of natural gas, which can help expand the annual production capacity of methane.

2.3. Supporting data of technology readiness

Table R4. Comparison of the bunkering technology readiness level (TRL) of different fuels for maritime sector (Level 9 represents the highest TRL)

Fuel type	Bunkering (Equipment) [16]	Bunkering (Fuel quality standards) [16]
Fuel oil	9	9
Hydrogen	4	3
Ammonia	7	2
Methanol	7	3
Methane	9	9

Table R5. The maturation timeline for engine technology for different fuels

Fuel type	Maturation for 2-Stroke Engine (TRL 9) [3]	Maturation for 4-Stroke Engine (TRL 9) [3]
Fuel oil	Already	Already
Hydrogen	No timeline	2028
Ammonia	2025	2026
Methanol	2022	2024
Methane	Already	Already

2.4. Supporting data of safety

Table R6. Comparison of the flash point of different fuels for maritime sector (The value of ammonia is used as the baseline, and Level 10 represents the highest level of safety)

Fuel type	Flash point (K) [35]	Relative value (%)	Level
Fuel oil	333	82.0	9
Hydrogen	20	4.9	1
Ammonia	405	100.0	10
Methanol	284	70.1	8
Methane	85	21.0	3

Table R7. Comparison of the minimum volume/mass flammability limit in air of different fuels for maritime sector (The value of ammonia is used as the baseline, and Level 10 represents the highest level of safety)

Fuel type	Minimum volume [7] / mass flammability limit in air (%)	Relative value (%)	Level
Fuel oil	1.0 / ^a 3.4	6.7 / 35.9	1 / 4
Hydrogen	4.0 / ^a 0.3	26.7 / 3.1	3 / 1
Ammonia	15.0 / ^a 9.4	100.0 / 100.0	10 / 10
Methanol	6.0 / ^a 6.6	40.0 / 70.2	4 / 8
Methane	5.0 / ^a 2.8	33.3 / 30.1	4 / 4

^a The minimum mass flammability limit in air is calculated from the minimum volume flammability limit in air from [7], and the relative molecular mass of air is assumed to be 29 in the calculation processes.

Table R8. Comparison of the toxicity of different fuels for maritime sector

Fuel type	Toxicity [36]
Fuel oil	Non-toxic
Hydrogen	Non-toxic
Ammonia	Toxic
Methanol	Toxic
Methane	Non-toxic

Table R9. Comparison of the corrosiveness of different fuels for maritime sector

Fuel type	Corrosiveness [36]
Fuel oil	Non-corrosive
Hydrogen	Non-corrosive
Ammonia	Corrosive
Methanol	Corrosive
Methane	Non-corrosive

Table R10. The maturation timeline for safety regulation of on board use for different fuels

Fuel type	Maturation for safety regulation of on board use [3]
Fuel oil	Already
Hydrogen	2032
Ammonia	2028
Methanol	2022
Methane	Already

2.5. Supporting data of emissions

Table R11. Comparison of the engine-out NO_x, particulate matter (PM), CO, hydrocarbon (HC), formaldehyde (HCHO), SO_x, and N₂O emissions of different fuels.

		Fuel oil	Hydrogen	Ammonia	Methanol	Methane
NO _x	^b Level	High	Dependent	Dependent	Dependent	Dependent
	Reasons or Refs.	Baseline	[37]	[38]	[39]	[40]
PM	^b Level	High	None	None	Dependent	Dependent
	Reasons or Refs.	Baseline	No carbon element in fuel		[39]	[41]
CO	^b Level	High	None	None	High	High
	Reasons or Refs.	Baseline	No carbon element in fuel		[39]	[42]
HC	^b Level	High	None	None	High	High
	Reasons or Refs.	Baseline	No carbon element in fuel		[39]	[42]
HCHO	^c Level	Low	None	None	High	High
	Reasons or Refs.	Baseline	No carbon element in fuel		[43]	[42]
SO _x	^b Level	High	None	None	None	None
	Reasons or Refs.	Baseline	No sulfur element in fuel			
N ₂ O	^d Level	1 (<10%)	1 (<10%)	5 (43%)	1 (<10%)	1 (<10%)
	N₂O emission (g/kWh)	0.027 [12]	^a 0.000 [12]	1.000 [28]	^a 0.000 [12]	0.016 [12]

^a According to Ref. [12], the engine-out N₂O emissions are regarded as zero for hydrogen and

methanol engine. However, for a more rigorous evaluation, this value should be near zero but not exactly zero because that the combustion of hydrogen and methanol will inevitably produce N₂O.

^b For NO_x, PM, CO, HC, SO_x emissions, High level: the value equal to or higher than that from diesel engine; Low level: the value smaller than 10% from diesel engine; None: no such emissions; Dependent: not belong to the first three levels, and the value is various under different engine types and operating conditions.

^c For HCHO emissions, Low level: the value equal to or lower than that from diesel engine; High level: the value larger than 10 times from diesel engine; None: no such emissions; Dependent: not belong to the first three levels, and the value is various under different engine type and operating conditions

^d For N₂O emissions, the Well-to-Propeller GHG emissions of diesel engine using fuel oil is employed as the baseline to obtain the relative value.)

The supporting data for GHG emissions by WtP CO₂ and CH₄ has been given in **Table R1**, while **Table R11** shows the supporting data for engine-out NO_x, particulate matter (PM), CO, hydrocarbon (HC) and formaldehyde (HCHO), SO_x, N₂O emissions of different fuels.

3. Main conclusions

The present techno-economic study conduct a comprehensive comparison on various alternative fuels (i.e. hydrogen, ammonia, methanol, methane) from five perspectives: economy, fuel availability, technology readiness level, safety and emissions. In the economic study, all the costs related to the use of alternative fuels have been considered as comprehensively as possible, including the fuel cost, fuel storage cost, engine and ship cost, extra voyage cost caused by the sacrifice of cargo capacity, GHG penalty by Well-to-Propeller CO₂ and CH₄ emissions, and GHG penalty by N₂O emissions from the current ammonia engine technology. Main conclusions are summarized as follows:

(The following main conclusions are also provided in Pages 19-21 for your convenience.)

- In terms of economy, in 2020, the annual fuel-related cost of using Green fuels is significantly higher than that of Blue fuels. However, in 2050, the price advantage of Blue fuels will be offset by the reduced Green fuel price and GHG emission penalties to some extent, resulting in the annual fuel-related cost of using Blue fuels being close to that of using Green fuels. As a result, owing to the significant price advantage, the Blue fuels that feature around 80% reduction in WtP CO₂ and CH₄ emissions (i.e. near-zero emissions) will serve as an important transitional fuel between fossil fuels and Green fuels in the next 30 years. Owing to the highest fuel storage cost and extra voyage cost, the use of hydrogen will result in the highest annual fuel-related cost. Due to the most expensive fuel cost, the annual fuel-related cost of using methanol fuel ranks second, only after the hydrogen fuel. Since the use of ammonia fuel features the low fuel cost and fuel

storage cost as well as low extra voyage cost, its annual fuel-related cost is significantly lower than the other fuels in all fuel price scenarios and years, even the GHG penalty by N₂O emissions having been totally considered.

- The economic advantages of using ammonia fuel in the future maritime sector will be more attractive if the N₂O emissions can be addressed. Taking the low fuel price scenario as an example, if the N₂O emissions from ammonia combustion engines can be solved by disruptive engine technologies, the annual fuel-related cost of Blue ammonia will be 57%, 60%, 62%, 66% of the cost of Blue methanol in 2020, 2030, 2040 and 2050, respectively. For the Green fuel in 2050, the annual fuel-related cost of ammonia is 82% of the cost of methanol with the consideration of economic penalty from N₂O emissions, while this value is reduced to 64% if the N₂O emissions from current ammonia engines can be solved. Moreover, if the N₂O emissions from the current ammonia combustion engine can be addressed, in 2050, the global international ships will save around 149 billion USD per year in GHG penalty from N₂O emissions. The above discussion about N₂O emissions highlights the economic advantages of using ammonia fuel in the future maritime sector and the necessity of the relevant disruptive engine technologies.
- In terms of technology readiness, the use of methane fuel has the same bunkering and engine TRL with fuel oil. The technological maturity of marine methanol engines can be achieved (i.e. 2022~2024) approximately 2 years ahead of that of marine ammonia engines (i.e. 2025~2026), but they can be considered to achieve technological maturity (i.e. TRL 9) at the same time in terms of the net-zero target in 2050. Correspondingly, the fuel quality standards for methanol fuel (i.e. TRL 3) are also slightly ahead of that for ammonia fuel (i.e. TRL 2), while the TRL of bunkering equipment is the same for ammonia and methanol fuel (i.e. TRL 7). The use of hydrogen fuel has the lowest bunkering and engine TRL, and this is mainly due to the economic and safety concerns.
- Flash point, minimum flammability limit in air, toxicity and corrosiveness are considered in the safety evaluation of different fuels. Methane has similar safety characteristics to fuel oil except for its extremely low flash point, while the low flash point and low flammability limit of hydrogen pose challenges in terms of safety. The poor combustion characteristics of ammonia fuel is the largest challenge for engine technology, but it is actually an advantage from the perspective of fire hazard and safety. Although ammonia and methanol are toxic and corrosive, they have been safely transported by international ships as chemical cargoes for a long time, demonstrating that the safety management of these two fuels is feasible and mature. As a result, the necessary safety regulations of on board use are expected to be ready for ammonia fuel within 5 years.
- In terms of emissions, the Well-to-Propeller GHG emissions (CO₂ and CH₄) and engine-out N₂O emissions have been penalized and discussed in detail in the economic study. Due to the absence of carbon (C) and sulfur (S) elements, the use of hydrogen (H₂) and ammonia (NH₃) fuels will not produce engine-out emissions such as CO₂, SO_x, particulate matter (PM), CO, hydrocarbon

(HC) and formaldehyde (HCHO), making these two alternative fuels most attractive from a long-term perspective. Methanol (CH₃OH) and methane (CH₄), as carbon-containing fuels, exhibit similar engine-out emission characteristics (i.e. CO₂, PM, CO, HC, CH₄, formaldehyde etc.). Compared to the use of fuel oil, the use of methanol and methane can reduce PM emissions to some extent, but it will lead to a significant increase in unregulated harmful emissions like formaldehyde.

- As for the fuel availability, the global annual production of hydrogen is the highest and around 80% higher than that of ammonia, while the annual production of ammonia is approximately twice that of methanol. If all the considered alternative fuels were to be used in the maritime sector, they could meet approximately 78% of the total energy requirement. However, most of these alternative fuels produced to date are Gray and Brown fuels, highlighting the production capacity of Green and Blue fuels needs to be significantly increased in the near future to meet the IMO target of net-zero emissions in 2050.
- In summary, ammonia is the most competitive alternative fuel in a comprehensive consideration of economy, emissions, fuel availability, technology and safety. Especially, the annual fuel-related cost of using ammonia fuel is significantly lower than that of using the other fuels in all fuel price scenarios and years, even the GHG penalty by N₂O emissions having been totally considered. Compared to carbon-containing fuels like methanol and methane, the combustion of carbon-free ammonia fuel produces no engine-out emissions such as CO₂, PM, CO, HC, CH₄, formaldehyde etc., and the synthesis of ammonia does not require the expensive and immature direct air capture processes for obtaining CO₂, which corresponds to the low fuel production prices. Moreover, the use of ammonia fuel in maritime sector need not specially establish and monitor the complex carbon value chain compared to the use of methanol and methane fuels. Currently, in addition to the shortage of Blue and Green sources that all alternative fuels face, the most significant challenge in using ammonia as a marine fuel lies in engine combustion technology, specifically achieving technological maturity as soon as possible (i.e. around 2025~2026) and further addressing N₂O emissions through combustion optimizations and disruptive technologies. As carbon-containing alternative fuels, methanol and methane exhibit similar economy and emission characteristics and can be compared together. For example, using methanol and methane can eliminate SO_x emissions and reduce PM emissions, but it is still inevitable to produce NO_x, CO and HC emissions as well as a large amount of unregulated harmful emissions like formaldehyde. The bunkering and engine TRL of methane are higher than those of methanol, and the transport of methane can share the mature infrastructure of natural gas, which can help expand the annual production capacity of methane. However, the methanol is superior than methane in terms of on board storage cost and methane slip. The present techno-economic study also specifically compare ammonia fuel and methanol fuel, and more details are summarized in **Fig. R11**.

Hydrogen is the cleanest fuel, but its application in maritime sector is currently limited by issues such as high fuel-related cost, low TRL and safety problems.

- According to the current global annual production capacity, even if all the Green and Blue alternative fuels were applicable in the maritime sector, they would still far from enough to meet the large demands of global international ships. As a result, although ammonia has been considered as the most promising alternative marine fuel according to the results of the present study, the maritime sector still needs an era of coexistence of multiple alternative fuels to mitigate the current severe shortage of Green and Blue fuels. Considering the fuel-related cost of global international ships in 2050 calculated to be between 635 billion USD (i.e. the use of Blue ammonia in the low fuel price scenario) and 1,597 billion USD (i.e. the use of Green hydrogen in the high fuel price scenario) per year, any alternative fuel, even if it holds a small market share, is of significant academic and economic value. The above highlights the necessity and urgency of developing disruptive engine technologies to improve fuel efficiency and reduce emissions.

4. References

- [1] Engine retrofit report 2023: Applying alternative fuels to existing ships. Lloyd's Register Group Limited; 2023.
- [2] The 80nd session of its Marine Environment Protection Committee (MEPC 80). International Maritime Organization. July 2023.
- [3] Maritime forecast to 2050, Energy transition outlook 2022. Det Norske Veritas; 2022.
- [4] Ammonia as a marine fuel. Maersk Mc-Kineey Moller Center; February 2022.
- [5] Sustainability whitepaper, ammonia as marine fuel. American Bureau of Shipping; October 2020.
- [6] Ammonia as a marine fuel. Maritime Energy & Sustainable Development (MESD) Center of Excellence supported by Singapore Maritime Institute; October 2022.
- [7] Sustainability whitepaper, hydrogen as marine fuel. American Bureau of Shipping; June 2021.
- [8] Zero carbon outlook, setting the course to low carbon shipping. American Bureau of Shipping; 2022.
- [9] Alternative fuels for international shipping. Maritime Energy & Sustainable Development (MESD) Center of Excellence supported by Singapore Maritime Institute; 2020.
- [10] Methanol as a marine fuel. Maersk Mc-Kineey Moller Center; February 2022.
- [11] Sustainability whitepaper, methanol as marine fuel. American Bureau of Shipping; February 2021.
- [12] Methanol as a marine fuel. Maritime Energy & Sustainable Development (MESD) Center of Excellence supported by Singapore Maritime Institute; January 2021.
- [13] Methane as a marine fuel. Maersk Mc-Kineey Moller Center; February 2022.

- [14] Fuel for thought: Methanol. Lloyd's Register Group Limited; 2023.
- [15] Industry transition strategy: we show the world it is possible. Maersk Mc-Kinney Moller Center; October 2021.
- [16] Techno-economic assessment of zero-carbon fuels. Lloyd's Register Group Limited; March 2020.
- [17] Bio-oils as marine fuel. Maersk Mc-Kinney Moller Center; February 2022.
- [18] Sustainability whitepaper, biofuels as marine fuel. American Bureau of Shipping; May 2021.
- [19] A pathway to decarbonise the shipping sector by 2050. International Renewable Energy Agency (IRENA); 2021.
- [20] Maritime forecast to 2050, Energy transition outlook 2023. Det Norske Veritas; 2023.
- [21] Carbon capture, utilization and storage. American Bureau of Shipping; August 2021.
- [22] Insights into onboard carbon capture. American Bureau of Shipping; 2022.
- [23] Onboard carbon capture utilisation and storage. Lloyd's Register Group Limited; April 2023.
- [24] Position paper: fuel option scenarios. Maersk Mc-Kinney Moller Center; October 2021.
- [25] Fuel production cost estimates and assumptions. Lloyd's Register Group Limited; January 2019.
- [26] Zero-emission vessels: transition pathways. Lloyd's Register Group Limited; January 2019.
- [27] Technology roadmap: hydrogen and fuel cells. International Energy Agency (IEA); 2015.
- [28] Wolfram P, Kyle P, Zhang X, Gkantonas S, Smith S. Using ammonia as a shipping fuel could disturb the nitrogen cycle. *Nature Energy* 2022;7:1112-14.
- [29] Zhou X, Li T, Wang N, Wang X, Chen R, Li S. Pilot diesel-ignited ammonia dual fuel low-speed marine engines: A comparative analysis of ammonia premixed and high-pressure spray combustion modes with CFD simulation. *Renewable and Sustainable Energy Reviews* 2023;173:113108.
- [30] Tripathi S, Gorbatenko I, Garcia A, Sarathy M. Sustainability of future shipping fuels: Well-to-Wake environmental and techno-economic analysis of ammonia and methanol. SAE paper 2023-24-0093; 2023.
- [31] Sollai S, Porcu A, Tola V, Ferrara F, Pettinau A. Renewable methanol production from green hydrogen and captured CO₂: A techno-economic assessment. *Journal of CO₂ Utilization* 2023;68:102345.
- [32] Stolz B, Held M, Georges G, Boulouchos K. Techno-economic analysis of renewable fuels for ships carrying bulk cargo in Europe. *Nature Energy* 2022;7:203-12.
- [33] McKinlay CJ, Turnock SR, Hudson DA. Route to zero emission shipping: Hydrogen, ammonia or methanol? *Int J Hydrog Energy* 2021;46:28282-97.
- [34] Losson L. Global biomethane market 2023 assessment - from ambition to action. *Biogas-Biomethane-Green Gas*; 20 April 2023.

- [35] Pathways to sustainable shipping. American Bureau of Shipping; 2020.
- [36] Law LC, Foscoli B, Mastorakos E, Evans S. A comparison of alternative fuels for shipping in terms of lifecycle energy and cost. *Energies* 2021;14:8502.
- [37] Ma D, Sun Z. Progress on the studies about NO_x emission in PFI-H₂ICE. *Int J Hydrog Energy* 2020;45:10580-91.
- [38] Mi S, Wu H, Liu C, Zheng L, Zhao W, Qian Y, et al. Potential of ammonia energy fraction and diesel pilot-injection strategy on improving combustion and emission performance in an ammonia-diesel dual fuel engine. *Fuel* 2023;343:127889.
- [39] Liu H, Zhang X, Zhang Z, Wu Y, Wang C, Chang W, Zheng Z, Yao M. Effects of 2-ethylhexyl nitrate (EHN) on combustion and emissions on a compression ignition engine fueling high-pressure direct-injection pure methanol fuel. *Fuel* 2023;341:127684.
- [40] Hlaing P, Marquez ME, Cenker E, Im HG, Johansson B, Turner JWG. Effects of volume and nozzle area in narrow-throat spark-ignited pre-chamber combustion engines. *Fuel* 2022;313:123029.
- [41] Corbin JC, Peng W, Yang J, Sommer DE, Trivanovic U, Kirchen P, Miller JW, Rogak S, et al. Characterization of particulate matter emitted by a marine engine operated with liquefied natural gas and diesel fuels. *Atmospheric Environment* 2020;220:117030.
- [42] Peng W, Yang J, Corbin JC, Trivanovic U, Lobo P, Kirchen P, Rogak S, Gagne S, et al. Comprehensive analysis of the air quality impacts of switching a marine vessel from diesel fuel to natural gas. *Environmental Pollution* 2020;266:115404.
- [43] Zhang Y, Mu Z, Wei Y, Zhu Z, Du R, Liu S. Comprehensive study on unregulated emissions of heavy-duty SI pure methanol engine with EGR. *Fuel* 2022;320:123974.

5. Nomenclature

ABS	American Bureau of Shipping
BECCS	bioenergy with carbon capture and storage
DAC	direct air capture
DME	dimethyl ether
DNV	Det Norske Veritas
DWT	dead weight tonnage
FAME	fatty-acid methyl ester
FT	Fischer-Tropsch
GHG	greenhouse gas
IEA	International Energy Agency
IMO	International Maritime Organization

LR	Lloyd's Register
LSFO	low-sulphur fuel oil
PM	particulate matter
SMR	steam methane reforming
SVO	straight vegetable oil
TRL	technology readiness level
WtP	Well-to-Propeller

5. To Reviewer #3: Point-by-point response

This article has a state-of-the-art results for ammonia-diesel multi cylinder engine operation with dedicated EGR concept. 3D CFD analysis was used after proper validation process of real engine operation. The results have a good quality and are timely appropriate on demand of decarbonization for maritime propulsion. I think that after the revision process corresponding to some details, this article can be accepted to this journal.

Re: We very appreciate your kind comments and encouragement.

1. page 5, line 118

: As far as I know, partial oxidation of ammonia, under deficient amount of oxygen smaller than that of stoichiometry, has low yield of hydrogen compared to ammonia decomposition itself. The author should clarify the reason why partial oxidation was used rather than decomposition - one of the expected reasons is that to run the entire engine successfully, exothermic reaction should be required for every cylinder.

Re: Thank you for your kind reminder. Sorry for the inaccurate expression. We have changed the expression of "*partially reformed*" to "*partially decomposed*" in line 22 and line 242 of the revised manuscript according to your valuable suggestion.

2. page 5, line 126

: I'm afraid but there is no citation for some literatures related to the concept of dedicated EGR which is similar to IRGR. Although the application of 'dedicated EGR' to ammonia-fueled engine has its novelty as first attempt, the author should clearly mention the history of dedicated EGR and what the difference is between that and IRGR in this study.

Re: Thank you for your kind and invaluable suggestion, we have added the history of dedicated EGR and what the difference is between that and IRGR in lines 248-271 of the revised manuscript, the relevant text in the revised manuscript is as follows:

"It should be noted that a comparable technology termed as dedicated EGR was proposed by Alger et al. [44] in 2007 to improve the EGR tolerance of a spark-ignition gasoline engine. The initial version of dedicated EGR [44] includes a water-gas shift catalyst in the exhaust manifold to promote the water-gas shift reaction (i.e. $CO + H_2O = CO_2 + H_2$), it was found that a small amount of hydrogen (i.e. around 0.2% by volume) added in the intake manifold is helpful for improving the combustion efficiency and engine performance. Two years later, Alger et al. [45] proposed a modified version of dedicated EGR in 2009, one of the main modifications is the use of partial oxidation catalyst (PO_x) in the exhaust runner of dedicated cylinder to convert some of the unburned hydrocarbons (HC) in the exhaust to CO and H_2 . They reported that the dedicated EGR is useful for reducing the fuel

consumption and improving the engine stability. In 2014, Chadwell et al. [46] further improved the dedicated EGR technology through optimizing the relevant boosting, EGR control, EGR mixing and ignition systems, and obtained the performance map of a 2.0 L gasoline direct injection engine. They demonstrated that the dedicated EGR technology can improve the engine efficiency by around 10%. Later, Gukelberger et al. [47, 48] further studied the potential of applying dedicated EGR to a gasoline engine and evaluated the engine performance under various engine loads. They demonstrated that the engine with dedicated EGR has the ability to operate at high compression ratio and therefore has higher thermal efficiency. Moreover, they highlighted that a water-gas shift catalyst added in the exhaust runner of dedicated cylinder can promote the water-gas shift reaction and increase the EGR quality. In recent years, there have also been some studies demonstrating that the dedicated EGR can improve the dilution tolerance of spark-ignition natural gas engine [49]. Compared to the above dedicated EGR technology proposed for improving the EGR tolerance of spark-ignition engine, the IRGR pilot-diesel-ignition ammonia combustion engine in the present study exhibits differences in terms of the independence of Cylinder #1's intake system (i.e. independent for IRGR vs shared with other cylinders for dedicated EGR), whether the EGR catalyst is suggested, ignition and combustion mode (i.e. diffusion and premixed combustion vs only premixed combustion), fuels (i.e. ammonia for IRGR vs carbonaceous fuels like gasoline and natural gas for dedicated EGR) and so on. To the authors' best understanding, so far there has been no study to verify the potential of similar concept applied to ammonia-fueled engines."

3. page 5, line 132:

: It was mentioned that experiments were conducted without IRGR which was numerically considered. Thus, the author should mention some possible problems that might happen in real operation of IRGR such as unbalanced torque distribution and vibration, backfire of reformed hydrogen over certain fraction.

Re: Thank you a lot for your insight suggestions. Some of the relevant issues have been highlighted in our previous manuscript as below: "*Since the total input energy of the dedicated reforming cylinder is higher than that of the other cylinders, the IMEP of the dedicated reforming cylinder will be slightly higher, but all the cylinders of the IRGR engine maintain the IMEP of around 11.4 ($\pm 10\%$) bar at the best thermal efficiency point. It should be noted that the IMEP difference between the dedicated reforming cylinder and Cylinders #2-4 can be alleviated by methods such as controlling the diesel injection timing and injection mass. However, as mentioned above, the control parameters such as diesel injection timing of the dedicated reforming cylinder are not optimized and directly set at -6°CA aTDC to avoid the unacceptable computational burden and highlight the effectiveness of the IRGR concept.*" In addition, the following issues are highlighted in lines 395-397 of the revised manuscript

according to your kind suggestion: "*For the real operation of IRGR ammonia combustion engine, while the above IMEP difference is acceptable in terms of engine balance and vibration, it should be minimized as much as possible. For example, the IMEP difference between the dedicated reforming cylinder and Cylinders #2-4 can be alleviated by methods such as...*". Moreover, as for backfire issue, we think it can be avoided from the experience of hydrogen engine design, and we have highlighted this information in lines 456-457 of the revised manuscript: "*It should be noted that for the real operation of IRGR ammonia combustion engine, it is necessary to refer to the experiences of hydrogen engine design to avoid the occurrence of backfire*". Thank you for your kind reminder again.

4. page 6, line 140

: I'm afraid but the readability of figure 2 is not well due to the large number of sub-figures. I would suggest to split figure 2 into three pieces - figure 2a, figure 2b, figure 2cdef, and figure 2ghi. If there is maximum number of figures in guideline, please ignore this comment.

Re: As suggested, we have divided Figure 2 into four pieces to improve the readability according to your valuable suggestions, as shown in Figs. 4, 5, 6, 7 of the revised manuscript, or **Fig. R2, Fig. R3, Fig. R4** in section "*Validation of 3D-CFD simulations against more experiment data*" in Pages 6-12 of this response file.

5. page 8, Table 2

: Considering the mention related to the comparison between No.6 and 7 in Table 2, should the number of reaction in No.6 be 341, rather than 344?

Re: No. 6 also includes these three reactions, but the reaction constants of these three reactions are obtained from other more recent literatures. Therefore, the number of reaction in No.6 is also 344. In order to avoid misunderstanding, we have used the following text in line 328 of the revised manuscript: "*For the newly developed mechanism (i.e. No. 7), it has the same 344 reactions as No. 6, but the reaction constants of $NH_3 + OH = NH_2 + H_2O$, $NH_2 + NO_2 = H_2NO + NO$ and $NH_2 + NO_2 = N_2O + H_2O$ have been updated based on the latest literatures. Specifically, the reaction constants for the first two reactions were obtained from [37], while the reaction constant for the third reaction is obtained from [61].*"

6. page 9, line 208

: As far as I understand, newly-added three elementary reactions for No.7 in Table 2, as mentioned in the last part of section 2.2, have an important role to satisfy the matching process between numerical and experimental data. Please clarify the reason otherwise the author should mention that it is beyond the scope of this work.

Re: Yes, the newly-modified three elementary reactions from more recent literatures play an

important role to satisfy the calibrating processes. Before developing mechanism No. 7, we first compared the effectiveness of mechanisms No. 1 to No. 6. We found that the mechanism No. 6 performed relatively well but still can not predict the experimental unburned ammonia data when ammonia energetic ratio is larger than 60%, as shown in Fig. 7 of the revised manuscript. As a result, we updated three elementary reactions closely related to ammonia oxidation based on the more recent literatures to satisfy the matching process between numerical simulations and experimental data. We have added the above information in line 332 of the revised manuscript according to your kind suggestion: *"The reason for the above modification is that the mechanism No. 6 performed relatively well among the first six mechanisms, but it still can not predict the experimental data of combustion phasing and unburned ammonia when ammonia energetic ratio is larger than 60%, as shown in Fig. 6 and Fig. 7. As a result, we updated the above three elementary reactions closely related to ammonia oxidation based on the more recent literatures to satisfy the calibrating processes"*.

7. page 10, line 235

: The original excess air ratio is relatively low compared to that of general CI diesel engine. Please clarify the reason why this value was selected - for example, low exhaust gas temperature and/or combustion instability due to ammonia.

Re: Yes, this is because the poor combustion characteristics of ammonia. As a result, to make a fair comparison and highlight the potential of IRGR concept, we selected a proper excess air ratio for the base engine without IRGR. We have added more description in line 404 of the revised manuscript: *"It should be noted that the selected overall excess air ratio of the base engine without IRGR is relatively low compared to that of general diesel engine, and this is because the poor combustion characteristics of ammonia fuel and the aim of fair comparison with the IRGR engine"*.

8. page 11, line 257

: I'd like to suggest using mole fraction rather than mass fraction because 0.8 % of hydrogen mass fraction can be considerable as considering its extremely low molecular level.

Re: We have used mole fraction in Fig. 8 of the revised manuscript or **Fig. R12** here according to your kind suggestion.

Fig. R12. (a) In-cylinder temperature and pressure as well as heat release rate, (b) in-cylinder ammonia and hydrogen mass as well as hydrogen conversion ratio, (c) in-cylinder NO and N₂O, (d) development of high-temperature region and hydrogen-rich region as well as in-cylinder temperature distributions of the dedicated reforming cylinder under various overall excess air ratios. In (b), the hydrogen conversion ratio is calculated as the ratio of in-cylinder hydrogen energy and total input ammonia energy. In (d), the red, green and cyan isosurfaces indicate the temperature of 2000 K, hydrogen mole fraction of 4.5% and 9.0%, respectively. (The followings remain consistent across all cases: 1000 rpm, 10% diesel energetic ratio, -6 $^{\circ}$ CA aTDC diesel injection and 318 K intake temperature).

REVIEWER COMMENTS

Reviewer #1 (Remarks to the Author):

My comments have been satisfactorily addressed. Proceed to publish.

Reviewer #3 (Remarks to the Author):

Thanks for your effort to revise the manuscript and response to the comments. I think that the revised one has been improved much better as considering the comments directly. Therefore, I would like to suggest this one be accepted for the journal.

Reviewer #4 (Remarks to the Author):

The manuscript is interesting and very relevant considering that the shipping sector is eagerly looking to ammonia as a potential marine fuel and there are concerns about the emissions of nitrogen-based compounds. The main purpose of the paper is to perform numerical simulations to predict the engine concept for ammonia. I understand that the techno-economic assessment is added as part of the reply to comments from one of the reviewers. However, I think the techno-economic assessment performed is not relevant to the purpose and scope of the study. Moreover, there is no novelty in the techno-economic assessment performed as there are several such studies published comparing different alternative fuels in the marine sector. Also, the method used in the assessment is very simplified compared to existing studies leaving questions on the results. This techno-economic assessment part is not worth publishing as such. My suggestion is to limit the techno-economic assessment only to the concepts introduced in the study and compare them to traditional ammonia and diesel cases. Below are specific comments on the techno-economic assessment part and this part is not acceptable as such as the result would change.

1. The underlying data is critical for the result and it is recommended to use peer-reviewed scientific data for such assessment. The present assessment are not based on such sources and are questionable from life cycle perspective. One example is that well-to-propeller data as stated in manuscript 'Table A4-1 summarizes and calculates the Well-to-Propeller GHG emissions (CO₂ and CH₄) of 7 alternative fuels in Fig. A4-1 based on the latest data from LR [16], Maersk [4, 13, 15] and ABS [8]. It can be seen that only the Green hydrogen, Green ammonia, Green methanol can be considered as net-zero alternative fuels from the perspective of Well-to-Propeller GHG emissions (CO₂ and CH₄).' These results are sensitive towards the assumption and I would recommend using life cycle assessment studies and comparison shall be done with similar assumptions. Some of the recommendations are [1-4] and there are more scientific articles. This is particularly important in your analysis as you are looking into well-to-propeller, not tank-to-propeller and you have stated you have ignored 0.7 gCO₂-eq/MJ.
2. It is written that the 'cost related to the use of alternative fuels have been considered as comprehensively as possible, including the fuel cost, fuel storage cost, engine and ship cost, extra voyage cost owing to the low energy density of the alternative fuels and the reduced cargo capacity, GHG penalty by Well-to-Propeller CO₂ and CH₄ emissions, and GHG penalty by N₂O emissions from the current ammonia engine technology, more details are summarized as follows:' Again the base assumptions considered are not from the peer-reviewed scientific article or no data collection methodology is adopted. For example, the method for extra voyage cost is very simple by simply comparing the energy density. Different ships have different complexities, for some ship types, volume would be critical and some others weight is critical. Considering extra voyage cost shall be based on the selected ship type. Weight is critical for the specific ship type considered in the study (bulk carrier). Underlying assumptions on the electricity cost, and natural gas cost for fuel cost calculation are also not clear, this is important as the total cost is sensitive to feedstock costs.
3. Most of the results in the study are adopted from non-scientific articles and are not calculated as part of the study. Adding this analysis doesn't add value to the manuscript (e.g. Fig 1 and Fig 2) and is not within the scope of the study including different time frames considered.

4. The N₂O emissions in the study are considered from different studies. Why in the calculation, the numerical simulation result from your study is not used?
 5. The use of pilot fuel is required for marine engines, why the emission due to the use of pilot fuel is not considered in the assessment?
 6. It is not clear how the fuel required is assumed, there is no mention of the efficiency or fuel consumption of engines for different fuels. For example, the efficiency changes with the heat of vaporization of the fuel, fuel slip (LNG and ammonia) etc...
- [1] M. Perčić, N. Vladimir, and A. Fan, "Life-cycle cost assessment of alternative marine fuels to reduce the carbon footprint in short-sea shipping: A case study of Croatia," *Applied Energy*, vol. 279, 2020, doi: 10.1016/j.apenergy.2020.115848.
- [2] S. S. Hwang et al., "Life Cycle Assessment of Alternative Ship Fuels for Coastal Ferry Operating in Republic of Korea," *Journal of Marine Science and Engineering*, vol. 8, no. 9, 2020, doi: 10.3390/jmse8090660.
- [3] F. M. Kanchiralla, S. Brynolf, E. Malmgren, J. Hansson, and M. Grahn, "Life-Cycle Assessment and Costing of Fuels and Propulsion Systems in Future Fossil-Free Shipping," *Environ Sci Technol*, vol. 56, no. 17, pp. 12517-12531, Sep 6 2022, doi: 10.1021/acs.est.2c03016.
- [4] F. M. Kanchiralla, S. Brynolf, T. Olsson, J. Ellis, J. Hansson, and M. Grahn, "How do variations in ship operation impact the techno-economic feasibility and environmental performance of fossil-free fuels? A life cycle study," *Applied Energy*, vol. 350, 2023, doi: 10.1016/j.apenergy.2023.121773.

Point-by-point response to the reviewers' comments @ NCOMMS-23-24260B-Z

To Reviewer #1:

My comments have been satisfactorily addressed. Proceed to publish.

Re: Thank you very much for your kind review and invaluable suggestions, which have significantly helped us improve the manuscript.

To Reviewer #3:

Thanks for your effort to revise the manuscript and response to the comments. I think that the revised one has been improved much better as considering the comments directly. Therefore, I would like to suggest this one be accepted for the journal.

Re: Thank you again for your invaluable time during the review process. Your detailed and professional suggestions are very helpful to us.

To Reviewer #4:

The manuscript is interesting and very relevant considering that the shipping sector is eagerly looking to ammonia as a potential marine fuel and there are concerns about the emissions of nitrogen-based compounds. The main purpose of the paper is to perform numerical simulations to predict the engine concept for ammonia. I understand that the techno-economic assessment is added as part of the reply to comments from one of the reviewers. However, I think the techno-economic assessment performed is not relevant to the purpose and scope of the study. Moreover, there is no novelty in the techno-economic assessment performed as there are several such studies published comparing different alternative fuels in the marine sector. Also, the method used in the assessment is very simplified compared to existing studies leaving questions on the results. This techno-economic assessment part is not worth publishing as such. My suggestion is to limit the techno-economic assessment only to the concepts introduced in the study and compare them to traditional ammonia and diesel cases. Below are specific comments on the techno-economic assessment part and this part is not acceptable as such as the result would change.

Re: Thank you very much for your kind review, understanding and encouragement. Your invaluable and professional comments are very inspiring for our further studies. We are very grateful for your observation that the techno-economic assessment is added in response to comments from one of the reviewers, and we could not agree more that the techno-economic assessment falls outside the scope of the present study, and most importantly, may dilute the focus of this work. We also agree that there have been several studies published comparing different alternative fuels in the marine sector. In order to avoid introducing further methodological issues and diluting the focus of this study, we have removed all the content related to the TEA from the manuscript (i.e. Figs. 1&2 and Appendix 4: Techno-economic study on various alternative marine fuels as well as the relevant references). Moreover, in the present study, we have thoroughly compared the new IRGR engine and traditional ammonia engine without IRGR from several perspectives such as thermal efficiency and nitrogen-based emissions, and there have been many studies comparing traditional ammonia engine and pure diesel engine over the past two years. As a result, once the fuel prices of Green/Blue ammonia under large-scale production are determined in the future, it will be easy to evaluate the economic and social benefits of the IRGR concept in comparison to traditional ammonia combustion engine and pure diesel engine. To highlight the focus of the present engine technology study, we have opted not to

conduct further economic analyses. We hope the above clarification are helpful and thank you again for your kind review and suggestions. The responses to your other invaluable and professional questions are given below:

1. The underlying data is critical for the result and it is recommended to use peer-reviewed scientific data for such assessment. The present assessment are not based on such sources and are questionable from life cycle perspective. One example is that well-to-propeller data as stated in manuscript ‘Table A4-1 summarizes and calculates the Well-to-Propeller GHG emissions (CO₂ and CH₄) of 7 alternative fuels in Fig. A4-1 based on the latest data from LR [16], Maersk [4, 13, 15] and ABS [8]. It can be seen that only the Green hydrogen, Green ammonia, Green methanol can be considered as net-zero alternative fuels from the perspective of Well-to-Propeller GHG emissions (CO₂ and CH₄).’ These results are sensitive towards the assumption and I would recommend using life cycle assessment studies and comparison shall be done with similar assumptions. Some of the recommendations are [1-4] and there are more scientific articles. This is particularly important in your analysis as you are looking into well-to-propeller, not tank-to-propeller and you have stated you have ignored 0.7 gCO₂-eq/MJ.

Re: We agree that there are various fuel production pathways and the results are sensitive towards the assumptions during the fuel production processes. As you know, although most of the alternative fuels produced to date are Gray and Brown fuels, we primarily focus on the Green E-fuels and Blue fuels summarized in Fig. 4-1. To maintain the objectivity as possible as we can, we summarized the latest data from renowned maritime organizations like Lloyd’s Register, Maersk and American Bureau of Shipping in Table A4-1, and then calculated the averaged value from these three organizations for further analysis. We greatly appreciate your sharing of four invaluable references, which provide peer-reviewed scientific data that we are likely to use in the near future. We find that the Ref. 4 you provided can support our opinion that ammonia is the most economical alternative fuels in most scenarios, and we have mentioned Ref. 4 in the Introduction of our revised manuscript. Moreover, in Ref. 4, the energy consumption of direct air capture for methanol synthesis is assumed to be 0.600-1.230 kWh/kgCO₂, which is significantly lower than the value used in LR’s fuel price calculation (i.e. 2.631 kWh/kgCO₂) and the value within our knowledge. As a result, the economic advantage of using ammonia fuel might be even more obvious. Finally, we want to clarify that we want to convey that 0.7 gCO₂-eq/MJ is negligibly small compared to the value of fuel oil (i.e. 91.1 gCO₂-eq/MJ), and therefore the Green E-ammonia, Green E-hydrogen and Green E-hydrogen can be considered as net-zero fuel. We will revise this statement in the future study according to your kind reminder.

2. It is written that the ‘cost related to the use of alternative fuels have been considered as comprehensively as possible, including the fuel cost, fuel storage cost, engine and ship cost, extra voyage cost owing to the low energy density of the alternative fuels and the reduced cargo capacity, GHG penalty by Well-to-Propeller CO₂ and CH₄ emissions, and GHG penalty by N₂O emissions from the current ammonia engine technology, more details are summarized as follows:’ Again the base assumptions considered are not from the peer-reviewed scientific article or no data collection methodology is adopted. For example, the method for extra voyage cost is very simple by simply comparing the energy density. Different ships have different complexities, for some ship types, volume would be critical and some others weight is critical. Considering extra voyage cost shall be based on the selected ship type. Weight is critical for the specific ship type considered in the study

(bulk carrier). Underlying assumptions on the electricity cost, and natural gas cost for fuel cost calculation are also not clear, this is important as the total cost is sensitive to feedstock costs.

Re: We totally agree with your comments above, and we have also mentioned similar issues in Appendix 4: *"It should be noted that when assessing the impact of alternative fuels on cargo capacity of bulk carrier, both the volume and weight of the fuel and fuel tank should be considered. In the calculation of the extra voyage cost, the present economic study primarily considers the loss of cargo volume capacity resulting from the use of alternative fuels, which is more relevant for other vessel types such as container ships"*. We agree that weight is critical for most of bulk carriers that carry heavy cargoes such as iron ore, meanwhile, to the best of our knowledge, volume would be critical if the bulk carrier carrying light cargoes such as wood. As a result, to avoid too long the Appendix 4, we primarily consider the loss of cargo volume, which is more relevant for other vessel types such as container ships. In the calculation of extra voyage cost, in addition to considering the energy density, we also consider the volume of fuel tank and additional equipment based on peer-reviewed scientific article. Thank you for your kind reminder, and we will consider both the volume and weight in our future calculations of extra voyage cost according to your kind suggestion.

As you kindly mentioned, the assumptions on the feedstock costs such as electricity cost and natural gas cost are important for evaluating the fuel price. In our current TEA, the fuel prices are obtained from Lloyd's Register (i.e. Ref. 16 in Appendix). During the calculation process, Lloyd's Register used the natural gas cost from World Bank and International Energy Agency (IEA) and electricity cost from International Renewable Energy Agency (IRENA). For example, for the high fuel price scenario, the electricity cost in 2020, 2030, 2040 and 2050 are 0.1 USD/kWh, 0.083 USD/kWh, 0.066 USD/kWh and 0.05 USD/kWh, respectively. When LR calculates the fuel prices of ammonia, hydrogen, methanol and methane, the electricity cost is kept consistent as you suggested. The fuel price of ammonia is significantly lower than that of methanol and methane, and this is mainly due to the fact that separating nitrogen from the air during the ammonia production process is more mature and cost-effective compared to Direct Air Capture (DAC) used in the fuel production processes of methanol and methane (The proportion of N₂ in the air is 78%, while the proportion of CO₂ is less than 0.1%). We agree that the feedstock costs will significantly impact the final fuel prices, and we think it is currently challenging to accurately evaluate the feedstock costs, especially for the CO₂ cost under the large-scale fuel production of Green E-methanol and Green E-methane. For example, American Bureau of Shipping and Lloyd's Register in maritime sector reported that only DAC and bioenergy with carbon capture (BECCS) can be recognized as reasonable sources of CO₂ for the Green E-methanol and Green E-methane synthesis processes (To the best of our knowledge, this definition is currently still under discussion). However, DAC and BECCS will significantly increase the fuel cost, introduce uncertainty, and reduce the reliability of fuel production and supply processes. Compared to the use of methanol and methane fuels, the use of ammonia fuel in maritime sector need not specially establish and monitor the complex carbon value chain, and this is one of the reasons why we have dedicated most of our efforts to the marine ammonia combustion engines in the recent years. Overall, there is still significant uncertainty regarding the future feedstock costs under large scale production, and this is also one of the main reasons for excluding the entire TEA from this study. We hope the above clarification are acceptable and thank you for your kind comment.

3. Most of the results in the study are adopted from non-scientific articles and are not calculated as part of the study. Adding this analysis doesn't add value to the manuscript (e.g. Fig 1 and Fig 2) and is not within the scope of the study including different time frames considered.

Re: We strongly agree that adding the TEA analysis is not within the scope of the present study. As a result, we have removed all the content related to the TEA from the manuscript (i.e. Figs. 1&2 and Appendix 4: Techno-economic study on various alternative marine fuels as well as the relevant references) according to your invaluable suggestion.

4. The N₂O emissions in the study are considered from different studies. Why in the calculation, the numerical simulation result from your study is not used?

Re: This is primarily due to the consideration of the manuscript's structure: the economic analysis is placed in the Introduction, before presenting our experiment and simulation results. Moreover, considering the variations in N₂O emissions under various engine bore diameters and operating conditions, we used a relatively high value of N₂O emissions from the existing literature (i.e. 1 g/kWh) to calculate the GHG penalty, and thereby highlighting the price advantage of using ammonia fuel.

5. The use of pilot fuel is required for marine engines, why the emission due to the use of pilot fuel is not considered in the assessment?

Re: Thank you for your kind reminder. This is mainly due to the fraction of pilot diesel is negligibly small and the consideration of future technology advancements. In the present study, the pilot diesel energetic ratio is as low as 3%, and this value will further reduce for the two-stroke low-speed marine main engine that features a higher in-cylinder temperature near the top dead center. For example, as you know, the pilot diesel fraction is around 2% for the current LNG marine main engines. However, we will consider the pilot diesel in the future TEA studies according to your kind suggestion.

6. It is not clear how the fuel required is assumed, there is no mention of the efficiency or fuel consumption of engines for different fuels. For example, the efficiency changes with the heat of vaporization of the fuel, fuel slip (LNG and ammonia) etc...

Re: In this TEA study, the engine thermal efficiency is kept consistent for all the alternative fuels, and this is also primarily due to the consideration of future technology advancements, where the thermal efficiency of alternative fuel engines will gradually approach that of pure diesel engines. We agree that currently, owing to issues such as fuel slip and poor combustion characteristics, the thermal efficiency of alternative fuel engines is generally lower than that of diesel engine. In the future TEA studies, we will consider the differences in fuel consumption between the alternative fuel engines and diesel engines according to your kind suggestion. Moreover, we will strive to provide more experimental data on alternative fuel marine engines in the near future, which can serve as valuable references for researchers conducting the TEA studies, thus collectively contributing to the achievement of the very urgent IMO GHG target. Thank you again for your professional comments from the perspective of economic analysis.

REVIEWERS' COMMENTS

Reviewer #4 (Remarks to the Author):

Thank you for the revised manuscript. By removing the added techno-economic assessment makes the manuscript more clear and focused. The reply from the authors are satisfactory. Therefore, I would recommend the manuscript for acceptance in the journal.